**RESEARCH**

# Cross-phyla protein annotation by structural prediction and alignment

Fabian Ruperti[1,2†], Nikolaos Papadopoulos[1,3†], Jacob M. Musser[1], Milot Mirdita[4], Martin Steinegger[4,5] and Detlev Arendt[1,6*]

†Fabian Ruperti and Nikolaos Papadopoulos contributed equally to this work and are co-first authors.

*Correspondence:
detlev.arendt@embl.de

[1] Developmental Biology Unit, European Molecular Biology Laboratory, Heidelberg, Germany
[2] Faculty of Biosciences, Collaboration for joint Ph.D. degree between EMBL and Heidelberg University, Heidelberg, Germany
[3] Department for Evolutionary Biology, University of Vienna, Vienna, Austria
[4] School of Biological Sciences, Seoul National University, Seoul, South Korea
[5] Artificial Intelligence Institute, Seoul National University, Seoul, South Korea
[6] Centre for Organismal Studies, University of Heidelberg, Heidelberg, Germany

## Abstract

**Background:** Protein annotation is a major goal in molecular biology, yet experimentally determined knowledge is typically limited to a few model organisms. In non-model species, the sequence-based prediction of gene orthology can be used to infer protein identity; however, this approach loses predictive power at longer evolutionary distances. Here we propose a workflow for protein annotation using structural similarity, exploiting the fact that similar protein structures often reflect homology and are more conserved than protein sequences.

**Results:** We propose a workflow of openly available tools for the functional annotation of proteins via structural similarity (**MorF**: *MorphologFinder*) and use it to annotate the complete proteome of a sponge. Sponges are highly relevant for inferring the early history of animals, yet their proteomes remain sparsely annotated. MorF accurately predicts the functions of proteins with known homology in >90% cases and annotates an additional 50% of the proteome beyond standard sequence-based methods. We uncover new functions for sponge cell types, including extensive FGF, TGF, and Ephrin signaling in sponge epithelia, and redox metabolism and control in myopeptidocytes. Notably, we also annotate genes specific to the enigmatic sponge mesocytes, proposing they function to digest cell walls.

**Conclusions:** Our work demonstrates that structural similarity is a powerful approach that complements and extends sequence similarity searches to identify homologous proteins over long evolutionary distances. We anticipate this will be a powerful approach that boosts discovery in numerous -omics datasets, especially for non-model organisms.

**Keywords:** Functional annotation, Proteins, *Spongilla*, scRNA-seq, Structural similarity, Structure, Conservation, Protein homology, Morpholog

## Background

Knowledge of protein function is crucial for interpreting many types of high-throughput molecular datasets. Since protein functional studies are limited to a few model species, amino acid sequence similarity has been used to predict the function of

protein homologs [1, 2]. However, homology detection over longer evolutionary distances remains challenging owing to the decay of protein sequence similarity that abolishes evidence of historical continuity. This presents a severe bottleneck for inferring protein function across a wide expanse of the tree of life, particularly in distant organisms where many proteins fall in the "twilight zone", only sharing a sequence identity between $10 - 20\%$ with proteins in characterized models [3, 4].

A way to venture more deeply into the twilight zone is to use structural similarity for homology detection, as structure is more conserved in evolution [5]. Until recently, this was not feasible since predicting protein structures from amino acid sequence required the prior existence of a homologous template structure [6]. This has changed with the advent of AlphaFold [7], a deep learning AI system that can predict de novo protein structures with atomic resolution, together with novel approaches for identifying similar structures in large databases [8]. Protein structures can now be predicted by AlphaFold for entire proteomes and then aligned to structures from model systems with characterized functions.

Sponges (*Porifera*) are animals that diverged early in the Metazoan tree relative to well annotated model organisms such as human and mouse. In this work, we predicted structures for the sparsely annotated proteome of the freshwater sponge *Spongilla lacustris* and aligned them against available structural databases to identify structurally similar proteins which we termed "morphologs" (from Greek *morphé* "form" and *lógos* "ratio"). We show that morphologs reflect homologous proteins in the vast majority of cases and often overlap in function even when homology is no longer detectable. We use morphologs to predict functions for unannotated sponge proteins by transferring functional annotations from model species. This complements sequence-based homology detection and subsequent function prediction. This expands the functional annotation coverage of the *Spongilla* proteome by 50%. Revisiting recent single-cell RNA-sequencing data [9], the novel annotations suggest additional aspects of sponge cell biology, such as extended cell signaling in pinacocytes, redox metabolism and control in myopeptidocytes, and polysaccharide digestion as a key function of the previously uncharacterized mesocytes.

## Results

### A protein structure-based workflow enriches functional annotation for *Spongilla lacustris*

We created a structure-based workflow for functional annotation transfer, which we refer to as MorF (*Mor*pholog*F*inder). Instead of using amino acid sequence similarity to assign homology and predict function, we predict protein structures, align them against structural databases, and transfer the functional annotation of the best hits, such as their preferred name and description, to the queries (for an overview, see Additional file 1: Fig. S1; details in the "Materials and methods" section). As a test case, we chose to annotate proteins in the freshwater demosponge *Spongilla lacustris*, an early-branching animal. With only about 20 cell types, organized into four cell families, *Spongilla* is a key model for understanding the origins of specialized animal cells [9].

We used the ColabFold [10] pipeline to predict three-dimensional structures for all 41,943 predicted *Spongilla* proteins, including isoforms (all structures and metadata deposited to ModelArchive [11], see the "Materials and methods" section). Eleven (11)

proteins were too long (>2,900 amino acids) to be predicted by the available hardware and were left unresolved. Confidence of predicted protein structures was assessed by calculating average pLDDT (predicted local distance difference) values. Average pLDDT values for *Spongilla* predicted protein structures were 4–6 percentage points lower than those of well-characterized animal models (Fig. 1A, [12]), likely reflecting the underrepresentation of sponges in the protein structure databases.

Next, we used the predicted protein structures as queries to search with Foldseek [8] against AlphaFoldDB [13], SwissProt [14], and PDB [15]. In general, bit scores for the best Foldseek hits were positively correlated with mean pLDDT. Neither parameter correlates with predicted physico-chemical properties of the proteins such as hydrophobicity, isoelectric points, or their instability index (Additional file 1: Section B, Additional file 1: Fig. S2-S4) [16]. After removing lower-quality matches, we retrieved functional annotations for the best morphologs using EggNOG-mapper (emapper) [17], a state-of-the-art orthology database [18], and then assigned these annotations to the protein in *Spongilla.* This produced annotations for slightly more than 60% of the proteome (25,232 proteins), representing an increase of approximately 50% compared to when *Spongilla* protein sequences were directly searched with emapper. Whereas the usage of emapper is not compulsory for MorF, it provided functional descriptors like EC numbers or GO terms for orthologous groups, facilitating later programmatic comparisons to sequence based methods. However, for downstream biological analysis, gene names and descriptions remain the most succinct, human-readable proxies for protein function. We therefore decided to use the preferred name and description of the best morpholog for each *Spongilla* protein assigned by emapper.

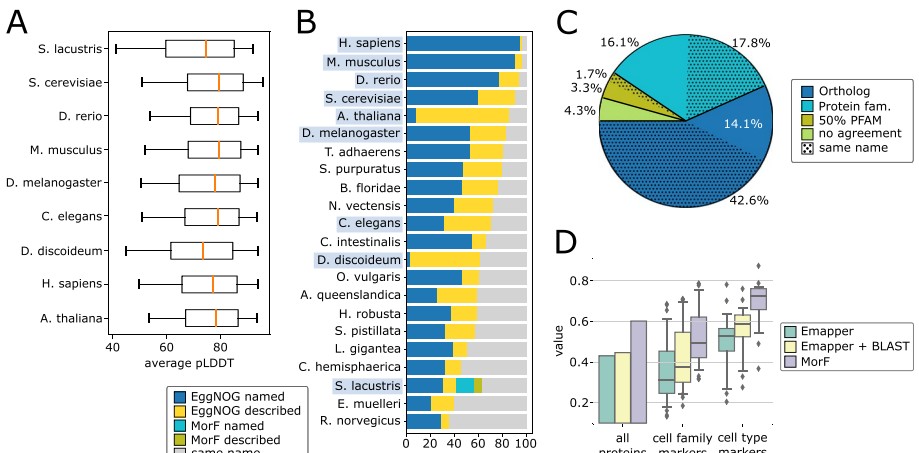

**Fig. 1** Structural prediction and alignment of the *Spongilla* proteome. **A** Distribution of average pLDDT for predicted proteomes from common model species in comparison to *Spongilla lacustris*. **B** Proportion of EggNOG or MorF protein annotations in *S. lacustris* and other eukaryotes. Highlighted organisms appear in **A**. **C** Overlap between EggNOG and MorF annotations. *Ortholog*: proteins identified as belonging to the same orthologous group in the most recent common ancestor in the EggNOG database. *Protein fam.*: proteins identified as belonging to the same eukaryote orthologous group in the EggNOG database, indicating annotations represent homologs in the same gene family. *50% PFAM*: half of the sequence-based PFAM domains are shared. *No agreement*: MorF and EggNOG annotations identify non-homologous proteins. Subcategories with "same name" denote the fractions where EggNOG and MorF returned the same preferred name for a protein. **D** Annotated proportions of different categories of *S. lacustris* proteins

We also compared our results to annotations from the recently published *Spongilla* cell type atlas, which used BLASTp to supplement emapper annotations [9]. We refer to these as "legacy annotations". Compared to this combined sequence-based approach, MorF annotates more proteins proteome-wide (∼60% to ∼40%). More importantly, MorF markedly improved the proportion of annotated cell type and cell family-specific markers (∼70%) compared with sequence-based approaches (∼56%; Fig. 1D, Additional file 1: Fig. S5), even considering sequence profiles (Additional file 1: Fig. S6 and S7).

### MorF annotations agree with sequence-based annotation transfer

Experimental evidence for the function of a protein is only available for a vanishingly small number of known protein sequences. For the rest, annotations are propagated, mostly using sufficient sequence similarity as a proxy for homology [19]. In particular, orthology has often been used to transfer functional annotation wholesale [1], even though it is known that divergence of function within orthologs is possible [20]. In the absence of robust high-throughput alternatives, sequence-based homology detection remains the standard for (transitive) functional annotation. To assess MorF annotations, we compare them to those produced via EggNOG v5.0 and EggNOG-mapper, representing the state-of-the-art in homology detection [17, 18].

We first examined how often the top non-trivial morpholog of a protein and the query protein itself belonged to the same family, using available predicted structures of model organisms. We aligned AlphaFoldDB against itself and kept for each query the top morpholog outside the species taxonomic unit. For Metazoa, we excluded all species belonging to the same phylum. Viridiplantae were divided into monocots and eudicots whereas fungi and trypanosome species were grouped by class. This ensured that MorF would not be identifying quasi identical proteins from closely related species (e.g., *Mus musculus* and *Rattus norvegicus*), simulating a realistic use case where MorF would be used to annotate a non-model organism without well-studied close relatives (Additional file 1: Table S3). We assessed performance by comparing the eukaryotic orthologous group of the top morpholog to that of the query protein, as defined by the EggNOG v5.0 database. MorF identifies 75–90% of all available homologs, indicating that it is largely able to reproduce sequence-based homology inference across large evolutionary distances.

We proceeded to repeat this analysis with the predicted *Spongilla* structures. A total of 16,589 proteins were annotated by both MorF and emapper. For 90.6% of these proteins, the MorF annotation was homologous to the EggNOG assignment (Fig. 1C), being either orthologs (56.7% in the same metazoan orthologous group) or in the same gene family (33.9% in the same orthologous group at the root level). Proteins that share the same gene family but are not annotated as orthologs either represent paralogs or have been misclassified, a problem for orthology inference that is prone to occur with large evolutionary distances [21]. In the remaining 9.4% of cases, approximately half shared a majority of their PFAM domains [22]. We explore the overlap between MorF and EggNOG in more detail in Additional file 1: Section G. Repeating this analysis with sequence profiles produced very similar results (Additional file 1: Section D).

### Morphologs share function over long evolutionary distances

As a next step, we sought to explore whether MorF can be used for functional annotation in cases where the evolutionary distance is too large for sequence-based approaches. To test this, we performed Foldseek searches for the predicted *S. cerevisiae* and *A. thaliana* proteomes against *Homo sapiens* and identified morphologs that lacked evidence of homology based on sequence. However, it is important to note that these cases could either represent homologs or protein structures that evolved convergently.

Nevertheless, for the remaining candidates, we tested their functional similarity by examining the overlap of their Enzyme Commission (EC) number where available. The EC number is a four-digit numerical description of enzyme function, with each number representing a progressively finer classification of the enzyme. Agreement on the first digit indicates two proteins are in the same broad enzyme class (oxireductases, hydrolases, ligases, etc.), while complete agreement means that they catalyze the same reaction.

For yeast, 109/145 (75%) enzymes agreed with their human morphologs on three of four EC positions and 53/145 (36.5%) agreed on all four. Similarly, for *Arabidopsis*, 357/532 eligible enzymes agreed to the third EC position (67%) and 176/532 (33%) had identical EC numbers. These results indicate MorF can accurately predict function even in cases where protein homology is unclear due to large evolutionary distances. Furthermore, this is consistent with other work in the field that has demonstrated that structure similarity uncovers homologs between *Homo sapiens* and different *Saccharomyces* species [23] using a similar methodology. We eagerly expect more insights on this topic in the coming months and years.

We next assessed the consistency of the functional annotation for top morphologs in *Spongilla*. For each protein, we queried the EC number of all morphologs in the 90th percentile of the Foldseek score range. This serves as an indirect way of validating that significant structural similarity correlates with functional conservation. In the 7072 cases that we could evaluate, the top morphologs were close to identical to the EC number of the best morpholog (average agreement 3.7 positions; see Additional file 1: Section H).

We also examined consistency between MorF and sequence-based annotations for *Spongilla* by comparing GO term overlap, semantic similarity and depth [24], drawing on the annotation comparisons of the CAFA challenge [25]. Almost all (99.9%) proteins annotated by both strategies have at least some overlap in GO terms with 60.6% being identical and 39.3% partially overlapping to various degrees (Additional file 1: Fig. S8A). GO term semantic similarities between proteins with overlapping (but non-identical) GO annotations additionally reach an average score of $80 - 90\%$ in the molecular function ontology (Additional file 1: Fig. S8B).

### MorF expands annotation of signaling pathways in sponge pinacocytes

A principal goal of functional annotations is to help identify cellular and molecular processes in large-scale genomic datasets. As a next step, we used MorF to revisit a recent single-cell RNA-sequencing dataset [9] which allowed us to confirm as well as expand the understanding of cell type functions in sponges. Musser et al. showed that sponge pinacocyte express members of the FGF, TGF/BMP, and Ephrin developmental signaling

pathways [26, 27]. Pinacocytes are contractile epithelial cells that line the sponge canal system, playing important roles in morphogenesis, barrier formation, and sponge whole-body contractions [28, 29]. Using MorF, we identify morphologs of additional members of these pathways expressed in pinacocytes, extending our understanding of their function in sponges.

In the FGF pathway, previous sequence-based annotations identified Fgf receptors and the FGF regulators Frs and Grb2, which are expressed in different pinacocyte cell types (Figure 2A). Extending this, MorF identified morphologs for GAB1 and GAB2 (GBR2-associated-binding protein 1/2) as well as PTPN11/SHP2, which are necessary for signal transmission into the cell [30, 31]. Notably, MorF also detected a morpholog of the FGF ligand in *Spongilla*, which was not found using sequence-based approaches. Structural superposition of the protein with its best Foldseek hit (UniProtID: P48804, *Gallus gallus* FGF4) revealed an extensive alignment of large parts of

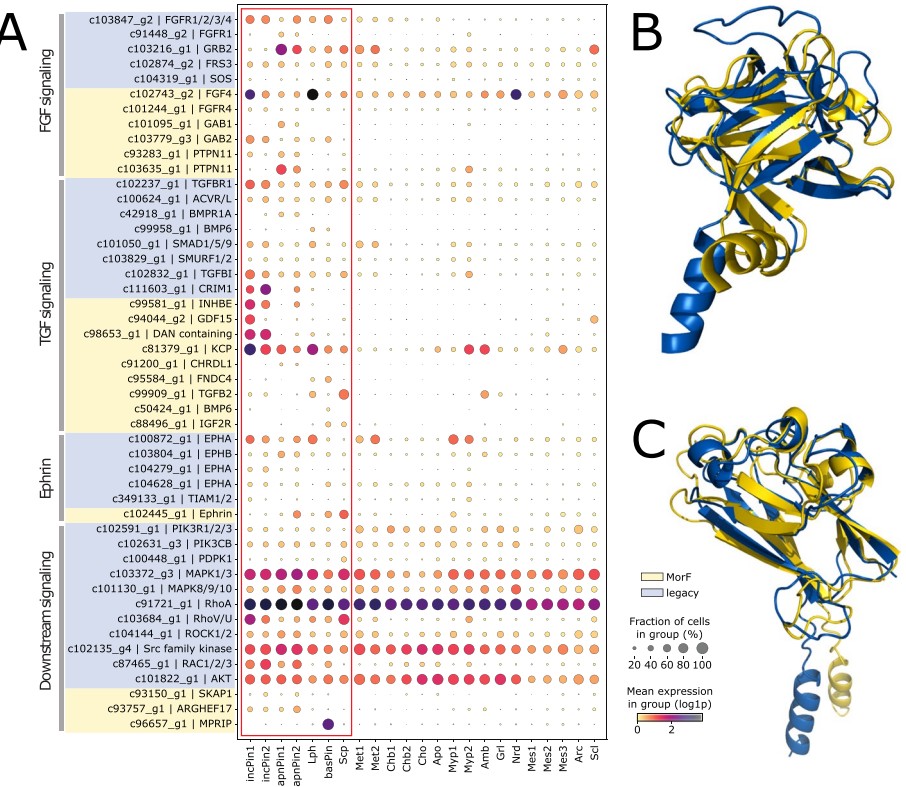

**Fig. 2** Signaling pathways in *Spongilla* pinacocytes. **A** Dotplot of pinacocyte signaling and effector genes. Cell types of the pinacocyte family are encased by a red square. Genes on blue background were annotated by Musser et al. with sequence-based methods ("legacy annotation"). Genes on yellow background are annotated by MorF. Showing mean expression of log-transformed, normalized counts. **B** Superposition of *Spongilla* FGF (blue, 61–230 aa) and *Gallus gallus* FGF4 (UniProtID: P48804) (yellow, 54–194 aa) (RMSD = 0.89 over 543 atoms). **C** Superposition of *Spongilla* ephrin (blue, 1–153 aa) and *C. elegans* efn-3 (UniProtID: Q19475) (yellow, 29–179 aa) (RMSD = 1.59 over 580 atoms). In both cases, the structural similarity is apparent despite their low sequence identity of 11.8% and 22% respectively. Superpositions were created using the `super` command in PyMOL (v2.3.5). Cell type abbreviations: incPin, incurrent pinacocytes; apnPn, apendopinacocytes; lph, lophocytes; basPin, basopinacocytes; scp, sclerophorocytes; met, metabolocytes; chb, choanoblasts; cho, choanocytes; apo, apopylar cells; myp, myopeptidocytes; amb, amoebocytes; grl, granulocytes; nrd, neuroid cells; mes, mesocytes; arc, archaeocytes; scl, sclerocytes

the protein, with a RMSD of 0.89 over 543 atoms despite a sequence identity of only 11.8% (Fig. 2B). Newly annotated players in the FGF pathway also exhibited enrichment in pinacocytes, consistent with its previously known cellular role.

Musser et al. also described multiple genes involved in TGF-$\beta$ signaling in the pinacocyte family, including Tgfbr1, Acvr, Smad, and Smurf [32, 33]. MorF extends the list of known actors in pinacocyte TGF signaling, adding morphologs of important ligands such as INHBE, CHRDL1, or KCP.

Lastly, Ephrin and the ephrin receptor (Eph) are membrane-anchored signaling molecules that mediate communication between adjacent cells [34, 35]. Whereas multiple ephrin receptors were detected via sequence similarity, ephrin itself was only found in the *Spongilla* proteome using highly sensitive HMM profile searches [36, 37]. MorF annotates a *Spongilla* gene with differential expression in various pinacocytes as a morpholog of *Caenorhabditis elegans* Efn-3 (UniProtID: Q19475). Although these proteins share only 22% sequence identity, the superimposed structures achieve an RMSD of 1.59 over 580 atoms (Fig. 2C). A separate Hmmer search [38] using the ephrin Pfam profile (PF00812) picked up the same gene, supporting MorF to be at least as sensitive as curated HMM profile searches (also see Additional file 1: Section D).

FGF, TGF-beta, and Ephrin interestingly exhibit converging downstream signaling pathways, including PI3K/Akt, ERK/MAPK1, JNK, or RhoA/ROCK, responsible for cell growth, differentiation, migration, and cytoskeletal organization [34, 39, 40]. Together, MorF and sequence-based methods identified morphologs of principal proteins involved in the downstream pathways. The genes encoding these proteins are broadly expressed across most *Spongilla* cell types, consistent with their diverse functional roles. This example highlights the power of MorF to vastly extend annotations and further elaborate cell type-specific functions.

### Redox metabolism and control in myopeptidocytes

The mesohyl of sponges is a collagenous, dynamic tissue forming large parts of the body between pinacocytes and the feeding choanocyte chambers [41]. Musser et al. identified five novel mesenchymal cell types [9] in *Spongilla*. Among them, myopeptidocytes are an abundant uncharacterized cell type, forming long projections that contact other cells. Sequence-based annotations suggested myopeptidocytes function to generate and degrade hydrogen peroxide by expressing dual oxidase (Duox1), its maturation factor (DuoxA), and Catalase (Cat) [42]. Myopeptidocytes also express transporters of copper ions as well as Ferric-chelate reductase (FRRS1 and FRRS1L), which recycle $Fe^{3+}$ to its reduced state, and suggest iron-based generation of $H_2O_2$. In their reduced state, metal ions react with $H_2O_2$ (Fenton reaction) [43] leading to the generation of hydroxyl radicals in cells. The existence of these prominent reactive oxygen species (ROS) (Fig. 3) is further supported by expression of Cyba [44]. However, further roles of ROS metabolism and function are unclear.

MorF predicted the functions of key additional members of ROS generation, metabolism, and response that are expressed in myopeptidocytes (Fig. 3, Additional file 1: Table S4). Morphologs of disulfide oxidoreductase (DsbB) as well as Flavin carrier protein (FLC), both playing a role in oxidative protein folding, have been detected by MorF [45, 46]. NmrA-like proteins act as redox sensors in the cell [47]. Consistent with a

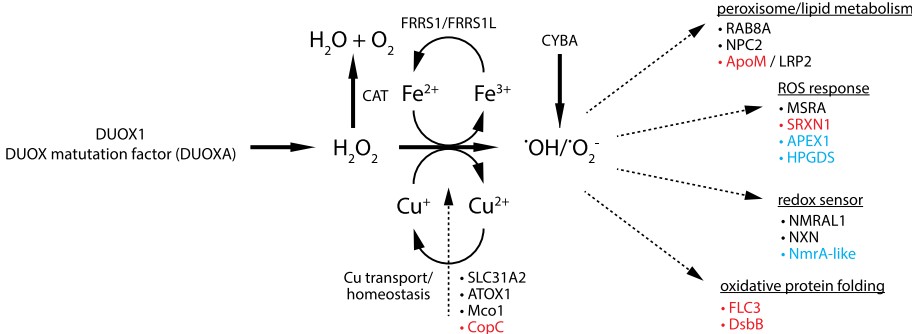

**Fig. 3** ROS metabolism and redox-control in myopeptidocytes. Myopeptidocytes differentially express multiple genes involved in redox control and ROS defense. Genes in black have been annotated using sequence based methods. Blue proteins have protein family level sequence-based annotation with updated functions inferred by MorF. Genes in red have been functionally annotated using MorF

possible redox regulation role, myopeptidocytes express morphologs of a range of additional ROS responsive proteins: Sulfiredoxin 1 (SRXN1) promotes resistance against oxidative stress damage [48], whereas AP endonuclease 1 (APEX1) protects against ROS-induced DNA damage [49, 50]. We also identified a glutathione S-transferase member (HPGDS) which together with previously annotated methionine sulfoxide reductase (msrA) is an important enzyme involved in the repair of proteins damaged by oxidative stress [51, 52]. Finally, myopeptidocytes express morphologs of the LDL-receptor LRP2 and its binding partner apolipoprotein M (ApoM) [53], suggesting a role in lipid metabolism and consistent with the observation myopeptidocytes exhibit round inclusions that may represent lipid droplets. Interestingly, lipid metabolism and redox control is tightly coupled in peroxisomes which are responsible for beta-oxidation of long-chain fatty acid [54]. Structure-based annotation of myopeptidocyte marker genes enabled us to substantially hypothesize about the role of this unexplored cell type in sponges.

## Polysaccharide hydrolysis in enigmatic mesocytes

Mesocytes are newly discovered medium-sized sponge cells whose name refers to their location in the mesenchymal mesohyl [9]. The single-cell RNA-seq data produced a series of marker genes specifically expressed in the mesocyte cell clusters; however, the lack of annotation for many of these genes made it difficult to hypothesize functions for these cell types.

New and refined annotations provided by MorF include proteins such as expansin (yoaJ), glucan endo-1,3-beta-glucosidase (BG3), and spore cortex-lytic enzyme (sleB), all hydrolases that specifically degrade cell walls, cellulase, chitin, other polysaccharides [55-58], and proteins (Table 1, Additional file 1: Table S5, Additional file 1: Fig. S9). "Refinement" here indicates that MorF provided names or descriptions for proteins previously annotated more sparsely, e.g., by a single predicted domain. It is thus tempting to speculate that mesocytes represent cells specialized to digest polysaccharides that are otherwise difficult to hydrolyse. For instance, the mesohyl has been shown to contain chitin which likely helps provide structural support to the sponge body [59]. The

**Table 1** Hydrolytic enzymes of *Spongilla* mesocyte marker genes

| Annotation | Annotation origin | Function | Putative gene origin |
| --- | --- | --- | --- |
| Aminohydrolase | legacy EggNOG | hydrolase acting on aminogroups | Bacteria |
| GMHA | legacy EggNOG | xanthan biosynthesis | Bacteria |
| Metallopeptidase M20 | legacy EggNOG | metallopeptidase | Bacteria |
| T5orf172 | legacy EggNOG | hydrolase activity | unknown |
| yoaJ | refined by MorF | cell wall degradation | Metazoa |
| sleB | refined by MorF | hydrolase activity, cell wall organization | Bacteria |
| BG3 | new MorF annotation | cellulase activity | Bacteria |
| cellulase A family member | new MorF annotation | cellulase activity | Bacteria |
| Endochitinase | new MorF annotation | endochitinase | Eukaryota |
| Chitinase class I | new MorF annotation | endochitinase | Eukaryota |

presence of chitinase in mesocytes suggests a possible role as structural remodelers of the sponge endoskeleton.

The sponge mesohyl also contains digestive and phagocytic cells [60] that process food particles captured by pinacocytes and choanocytes. These food particles often include bacteria and algae, which are protected by polysaccharide and glycoprotein cell walls [61] that require specialized enzymes to break down. While those enzymes are notably absent from the metazoan digestive repertoire [55], sponges are at least demonstrably able to digest algae [60].

The absence of these mesocyte-specific hydrolytic enzymes from the digestive toolkit of animals suggests four possibilities for their appearance in the *Spongilla* single-cell data: that they are an artifact (contamination), that they are an evolutionary novelty within *Porifera*, that they were lost in all other animal lineages, or that they were acquired via horizontal gene transfer (HGT).

To explore these different possibilities, we used the marker gene sequences to find putative homologs in the RefSeq non-redundant (*nr*) [62] and metagenomic databases. Strikingly, the best hits were mostly of bacterial origin, exhibiting $40 - 70\%$ shared sequence identity with sponge proteins; however, most lacked annotation (Additional file 1: Table 1). Notably, we identified putative homologs for each gene in other sponges, suggesting the presence of these sequences in the *Spongilla* protome is unlikely to have occurred due to contamination (Fig. 4B). Consistent with this, we found codon usage and GC content for these genes did not deviate from the *Spongilla* background (Fig. 4A). Lastly, we located all candidate genes on different long contigs (avg. length ~420kb) of an in-house draft assembly of the *Spongilla* genome. The specific co-expression of functionally similar proteins in mesocytes is in contrast to a random contamination.

The prospect of HGT is tantalizing. Proteins with enzymatic functions like the ones in the *Spongilla* candidates (polysaccharide hydrolases and metallopeptidases) have been proposed to be horizontally transferred in *Amphimedon queenslandica*, a marine demosponge; *Salpingoeca rosetta*, a choanoflagellate; and *Mnemiopsis leidyi*, a ctenophore [63-65]. Additionally, *Spongilla* genes c97022_g1 and c103983_g1, a putative aminohydrolase and metallopeptidase respectively, are not only broadly distributed within sponges, but can also be found in the proteomes of choanoflagellates *S. rosetta* and *Monosiga brevicollis* (Additional file 1: Section L). Furthermore, the *S. rosetta* targets with

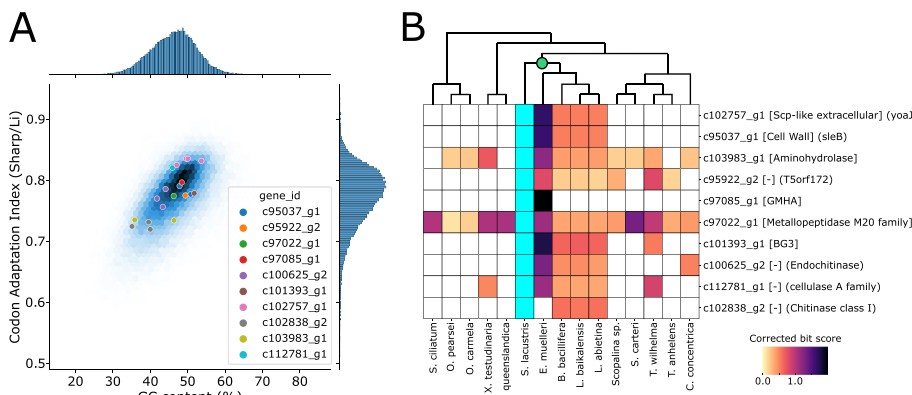

**Fig. 4** Horizontal gene transfer of mesocyte marker genes. **A** Distribution of GC-content and codon usage of mesocyte marker genes of non-metazoan origin (colored dots) compared to the entire *Spongilla* transcriptome (blue background). **B** Heatmap showing scores of best search hit of *Spongilla* mesocyte marker genes in various sponge species. The green dot denotes the last common freshwater sponge ancestor

highest similarity to the *Spongilla* sequences had already been identified as horizontally transferred genes [64]. This would tentatively place this HGT event at least before the split of choanoflagellates and animals (more than ∼500mya). Similarly, the phylogenetic distribution of c102757_g1, c95037_g1, and c102838_g2 (yoaJ-sleB-Chitinase class I) would indicate that this group of genes was acquired with the colonization of freshwater environments (∼15 − 300mya).

Although our analyses suggest these genes may have originated via HGT [66], it is important to consider alternative explanations. One possibility is that these genes are the results of widespread bacterial contamination in sponges genomes and proteomes. Other possibilities include the repeated loss of these genes in other animal lineages, or the convergent evolution of similarity with bacterial proteins. Additional confirmation of their presence in sponge genomes, or evidence of RNA transcripts in sponge cell nuclei, would help validate the hypothesis they arose via HGT. Regardless of the source of the genes, MorF annotations provided a novel hypothesis for the elusive function of sponge mesocytes, helping uncover new aspects of sponge biology.

### Identifying novel well-folded proteins in *Spongilla lacustris*

Using the MorF workflow with a stringent bit score cut-off and augmenting it with sequence-based annotations, we annotated a total of 26,633 out of 41,943 predicted proteins. The remaining 15,312 proteins may represent incomplete fragments, untranslated sequences, or sponge lineage-specific genes. Notably, we found 3875 unannotated proteins with a pLDDT score greater than 70, indicating well-folded structures. Although many of these had poor Foldseek alignments falling below our accepted bit score threshold, 316 had no Foldseek hit whatsoever. Manual inspection revealed that the overwhelming majority of these are predicted to be long helices, except for 35 non-helical structures. To ensure these had not somehow eluded sequence similarity searches, we used NCBI BLASTp to identify potential homologs in the NR database, even very remote ones (Additional file 1: Table S2). Seven of the sequences find no matches at any

theshold nor get significant PFAM domain hits. Notably, several of them are broadly expressed in *Spongilla* cell types, presenting prime candidates for truly novel structures.

Recent changes in the amount of available structures are almost certainly going to affect these numbers. The latest AlphaFoldDB version contains predicted structures for more than 214 million proteins [67], and with the continuous deposition of more sequences and their predicted structures our view of protein structure space will be ever closer to complete.

## Discussion

We predicted the protein structures of the entire proteome of the freshwater sponge *Spongilla lacustris* and aligned them against model organism proteomes, PDB, and SwissProt. This approach increased the annotation of the proteome from 40% to 60%, an approximately 50% increase. We found that in more than 90% of cases, sequence-based and structure-based annotations identified homologous proteins, a finding supported by recent work with well-annotated model species [23]. Additionally, these proteins overlap largely in their corresponding GO terms and show high semantic similarity.

The fact that morphologs are overwhelmingly homologs shows that structural and functional similarity between proteins mostly results from common descent. However, similar structures may, and do, also emerge convergently. MorF and sequence-based annotations disagree in ~5% of cases, representing either technical artifacts (Additional file 1: Section G), homology we simply cannot infer, or examples of convergent evolution. In the future, it will be interesting to explore experimentally whether structural similarity or homology is a better predictor of protein function [68].

To demonstrate the usefulness of MorF for functional annotation, we revisited the cell type marker genes previously identified from *Spongilla* scRNA-seq data [9]. In the epithelial pinacocyte family, we significantly improved annotation by detecting morphologs of key players in FGF, TGF-*β*, and Ephrin signaling. We were able to infer complex modes of redox regulation as a possible function of the myopeptidocytes. Finally, we detected polysaccharide and protein hydrolyzing enzymes potentially used for digesting cell walls in the so far enigmatic mesocytes, which may have originated from bacteria via HGT. If proven to be true, this example reveals cell types whose functional role is largely defined by genes acquired via HGT. A comparable example has been found in nematodes, which expanded their diet after the HGT of cellulase from a non-animal eukaryote [69]. These events are remarkable because they suggest HGT may help give rise to new cell type-specific functions [70]. Moreover, the large evolutionary distance between animals and bacteria has very likely hindered the identification of these events, and we anticipate MorF will uncover additional examples in other species. Recent evidence from nematodes showed an expansion of dietary possibilities after HGT of a eukaryotic cellulase gene. By re-evaluating cell type marker genes from the *Spongilla* cell type atlas, it became obvious that structure-based annotations not only expand the molecular context of known cell type functions but also allow substantiated hypotheses about previously unexplored cell types. These hypotheses can serve as a next step for further investigation and discoveries.

Well-folded *Spongilla* proteins without any annotation constitute intriguing candidates for novel protein folds and functions specific to sponges. Many of these proteins are predicted to be long alpha helices. This could either be an artifact of spuriously translated proteins (as described in [71]) or present sponge-specific proteins for structural integrity of the sponge body, similar to the "constant force spring" character of naturally occurring single $\alpha$-helices [72]. However, unannotated proteins with globular structures, expressed in the scRNA-seq data, are most probably functional and offer a great resource to study the evolutionary emergence of protein folds.

## Conclusions

The lack of reliable functional annotation has so far been a major bottleneck in the analysis of -omics datasets from non-model species, in particular those separated from traditional models by large phylogenetic distances. Here, we demonstrate that by exploiting the evolutionary conservation of protein structure it is possible to dramatically improve protein functional annotations in non-model species. We show that proteins with significantly similar structures (morphologs) are often homologs. Using GO-terms as well as EC numbers as measures for functional similarity, we illustrate that in many cases morphologs are functionally similar across large evolutionary distances and can therefore be used for functional transfer. Although protein structural predictions for an entire proteome might be outside the technical capabilities of many labs, the workflow described here can be used to query individual highly informative candidate genes from proteomics or single-cell -omics experiments. During the peer review process of this manuscript, protein structures for the entire UniProt database were predicted and updated version of Foldseek (v4) as well as the EggNOG database (v6.0) have been released. It is reasonable to expect that in the future protein sequences deposited in public databases will automatically receive predicted structures, paving the way for unique insights into biological functions across the tree of life.

## Materials and methods

In the manuscript, "annotation" of a protein refers to the existence of an emapper-based preferred gene name or description. In all boxplots, the box extends from the lower to upper quartile values of the data, with a line at the median. The whiskers represent the $5-95\%$ percentiles.

### Sequence-based annotation of the *Spongilla lacustris* proteome

Juvenile freshwater sponges (*Spongilla lacustris*), grown from gemmules, were used for bulk RNA isolation and sequencing. De novo transcriptome assembly with Trinity, returned 62,180 putative isoforms, covering 95.2% of Metazoan BUSCOs [73]. To identify putative proteins, Transdecoder [74] (version 3.0.1) was used with a minimum open reading frame length of 70 amino acids, resulting in 41,945 putative proteins. The longest putative protein per gene ID was kept. The resulting predicted proteome was annotated by EggNOG mapper [17, 18] (v2.1.7, default settings) via the website.

### *Legacy annotation*

Musser et al. [9] used the putative proteins to create a phylome by constructing gene/protein trees for each protein [75]. The phylome information was used to refine the assignment of transcripts to genes. In some cases, 3′ and 5′ fragments of a gene were assigned to two different transcripts. These fragments were merged into the same merged gene name using the gene tree information. Functional annotations were supplemented by EggNOG mapper (v1) and `blastp` searches against human RefSeq (default parameters). This annotation was used in the original *Spongilla* scRNA sequencing publication and is present in the single-cell data. We refer to this as "legacy annotation". In this manuscript, the legacy annotation was used for the single-cell data analysis, but not for the comparison between the MorF workflow and the sequence-based annotation transfer.

## Structure-based annotation of the *Spongilla lacustris* proteome

### *MorF and constituent tools*

We designed a simple workflow that goes from sequence to predicted structure to functional annotation (symbolically, MorphologFinder, or MorF), and used it to annotate the *Spongilla lacustris* predicted proteome. We used ColabFold to predict structures and subsequently aligned the predicted structures against all currently available (solved and predicted) protein structures using Foldseek. Finally, we used structural similarity to transfer annotations from the morphologs to their corresponding *Spongilla* protein queries. In the following we show an overview of the tools in use and a more detailed description of the MorF workflow.

*MMseqs2* (version 92deb92fb46583b4c68932111303d12dfa121364) [76] is a software suite for sequence-sequence and sequence-profile search and clustering. It is orders of magnitude faster than BLAST at the same sensitivity and is widely adopted [17].

*AlphaFold2* [7] is a neural network-based model that predicts protein three-dimensional structures from sequence, regularly achieving atomic accuracy even in cases where no similar structure is known. AlphaFold is widely considered to have revolutionized the field of structural bioinformatics, greatly outperforming the state of the art in the most recent iteration of the CASP challenge [77].

AlphaFold quantifies prediction confidence by pLDDT, the predicted local distance difference test on the $C\alpha$ atoms. Regions with pLDDT $> 90$ are modeled to high accuracy; regions with $70 < $ pLDDT $ < 90$ should have a generally good backbone prediction; regions with $50 < $ pLDDT $ < 70$ are low confidence and should be treated with caution; regions with pLDDT $< 50$ should not be interpreted and probably are disordered.

*ColabFold* [10] is a pipeline that combines fast homology searches via MMseqs2 [76] with AlphaFold2 [7] to predict protein structures 40 to 60 times faster than the original AlphaFold2. We installed `colabfold` locally from the `localcolabfold` repository [78], version 1.4.0

*Foldseek* [8] enables fast and sensitive comparison of large structure databases. Foldseek's key innovation lies in the appropriate translation of structure states to a small alphabet, thus gaining access to all the heuristics of sequence search algorithms. We used version 3-915ef7d.

### Structural databases

We used the *AlphaFold database* v1 [13], containing over 360,000 predicted structures from 21 model-organism proteomes, as provided by Foldseek v3-915ef7d.

From Foldseek v3-915ef7d, we also used the *Swiss-Prot* [6] and *PDB* [15] databases.

### The MorF workflow

A visual representation of the MorF workflow can be found in Supplement Fig. S1. To predict structures, we adapted the ColabFold pipeline as outlined in [79].

*Multiple sequence alignment generation*: We downloaded reference sequence databases (UniRef30, ColabFold DB) and calculated indices locally ([80, 81], adapted from ColabFold `setup_databases.sh`). We were interested in homology detection at the limit of the twilight zone, so UniRef30, a 30% sequence identity clustered database based on UniRef100 [82], was the adequate choice. We calculated MSAs for each *Spongilla* predicted protein ([83], adapted from [84]) using MMseqs2 [76].

*Structure prediction*: We predicted structures for all *Spongilla* predicted proteins using ColabFold [10] as a wrapper around AlphaFold2.

We split the MSAs in 32 batches and submitted each one to the EMBL cluster system (managed by `slurm` [85]); we used default arguments but added `-stop-at-score 85` [86]. The calculations were done on NVIDIA A100 GPUs, on computers running CentOS Linux 7. We used GCC [87] version 10.2.0 and CUDA version 11.1.1-GCC-10.2.0 [88]. We processed the resulting PDB-formatted model files with Biopython's PDB module [89].

*Structure search and annotation transfer*: Structural search was conducted using Foldseek which allows fast comparison of large structural databases. We downloaded PDB, SwissProt, and AlphaFold DB. For each *Spongilla* protein, we kept the best-scoring AlphaFold2 model, and used them to construct a Foldseek database. These models were then used to search against the three structural databases (see [90]). For each search, we kept the Foldseek hit with the highest corrected bit score in each database and aggregated the three result tables (AlphaFoldDB, PDB, SwissProt) into one. We imposed a bit score cutoff of $e^5$ on Foldseek hits based on their bimodal distribution [91] and personal communication with the Foldseek authors. Annotations of the best hits (= morphologs) were gathered from either UniProt via its API [92] or through EggNOG mapper (v2.1.7, default settings) [17] by using the sequences of the morphologs (pulled by UPIMAPI [93]). To facilitate downstream analysis, we extracted summary tables from each resource type. This procedure can be found in the corresponding notebook [94]. A total of 1401 proteins received sequence annotation but their Foldseek best hits were below the bit score cutoff.

### Instructions for MorF searches of single proteins using openly available web servers

MorF searches for whole proteomes require large computational resources. However, searches can be carried out for a small number of proteins of interest (e.g., top differentially expressed genes in RNAseq or proteomics experiments) using openly available web tools:

- Prediction of protein structure using ColabFold [10]: The structures of proteins of interest can be predicted using the ColabFold Google Colaboratory notebook [95]. Detailed instructions are described in the notebook. For a quick default run, users paste a protein sequence into "query_sequence" and hit "Runtime" - "Run all". The results can be downloaded as a zip archive which includes the pdb models of different structure model quality ranks.
- Structure similarity search using Foldseek [8]: The best ranking model ("...rank_1_model_X.pdb") can be queried using the Foldseek webserver [96]. Users can upload the pdb file of the model and select databases to use for the search. In default mode, all available databases will be searched. The Foldseek output is structured blast-like and sorted according to best scoring morphologs within the selected databases.
- (Optional) Additional annotation using EggNOG [17]: In order to retrieve additional functional as well as phylogenetic information about the best scoring morpholog, the EggNOG database can be searched [97]. Both protein sequence and UniProt ID can be used to retrieve information about orthologous groups as well as GO-terms, EC numbers, etc.

### GO term semantic similarity and depth calculation

For an in-depth comparison of sequence and structure-based annotations, we calculated the semantic similarity and depth of GO terms between annotation pairs [24]. Calculation of semantic similarities was done with the stand-alone version of GOGO [98] in default mode using the Average-Best-Match (ABM) method for calculating gene functional similarity [99, 100]. GO term semantic similarity was compared between all GO terms from annotation pairs with partially overlapping GO terms. Calculation of GO term depths was done with the GOATOOLS Python library [101]. GO term depths were compared using the overlapping GO term assignments between sequence- and structure-based annotations with partially overlapping GO terms [24].

### Differential gene expression in single-cell transcriptomics data

We obtained the processed Seurat file from [9, 102] and downloaded the lists of differentially expressed genes of clusters, cell types, and cell type clades from the supplemental material of the same publication (Suppl. Data S1 to S3; file science.abj2949_data_s1.xlsx; tabs "Diff. exp. 42 clusters", "Diff. Exp. cell types", "Cell type clade genes (OU tests)". The single-cell data operates on the level of genes, so we transformed the sequence-derived and MorF annotations by merging isoform entries and keeping the entry with the best bit score.

We used the legacy annotation included in the file (phylome, emapper, and BLASTp-based), the sequence-derived annotation, and the MorF output to propose names for *Spongilla* proteins. We prioritized sequence-derived annotations (legacy annotation, EggNOG preferred name, EggNOG description) and fell back to MorF (MorF preferred name, MorF description) when there were none. We produced dotplots for the top 200 differentially expressed genes in each cell type and manually inspected them. We focused on terminally differentiated (named) cell types; owing to the presence of continuously differentiating stem cells, the single-cell data contains many clusters of maturing

or differentiating cells whose expression patterns do not distinguish them from their mature counterparts. Code and detailed explanations are available in the corresponding notebook [103].

### Protein structure visualization and superposition

In order to visualize predicted structures from the *Spongilla* proteome, we used PyMOL Version 2.3.5 [104]. Superposition with their respective best Foldseek hit was carried out using the `super` command. `super` creates sequence-independent superpositions and is more reliable for protein pairs with low sequence similarity.

### Detection of ephrin orthologs in *Spongilla* using HMMER

To validate the ephrin ortholog detected by MorF in *Spongilla*, we recapitulated a previous effort to detect ephrins in different species using extensive HMMER protein profile search [37]. For this, we performed a HMMER search (v3.3.2) using the ephrin Pfam profile (PF00812) against the *Spongilla* proteome using default settings.

### Detecting HGT in *Spongilla*

To detect HGT, we followed the proposal by Degnan [66] and sought to get multiple indications for HGT. In particular: phylogenetic evidence that the candidate gene is more closely related to foreign than to animal genes; genome data showing the candidate gene assembles into a contiguous stretch of DNA with neighboring genes unambiguously of animal origin (this requires, of course, the availability of a sequenced and assembled animal genome; the more complete the assembly, the more confident the HGT identification); and gene sequence revealing metazoan-like compositional traits, including presence of introns, GC content, and codon usage. Where possible, gene expression data showing active transcription of candidate genes in animal cell nuclei can enormously strengthen a case, and also addresses the issue of whether or not the HGT-acquired gene is active in its new genomic context.

All putative HGT proteins were used as an input for default blastp searches against non-redundant protein (nr) as well as metagenome databases (env_nr). Additionally, default EggNOG 5.0 sequence searches were run. We used SpongeBase [105] to obtain transcriptomes and genomes for 13 sponge species and obtained a 14th one directly from its repository [106]. We built sequence databases with MMseqs2 version 12-113e3 and searched against them (`mmseqs easy-search`) with the protein sequences of all isoforms of the HGT candidates. The resulting alignments were filtered to keep the best-scoring alignment per species per gene.

The table of best hits per genome was visualized in Python, and a sponge phylogeny was added manually based on [107] (also A. Riesgo, personal communication).

## Supplementary Information

---

**Additional file 1.** Supplementary material [112–125].

**Additional file 2.** Review history.

---

### Acknowledgements

The authors thank Alexandros Pittis for his technical feedback; the Arendt lab for discussions; the ModelArchive team for their support in publishing the predicted structures; the EMBL Heidelberg High Performance Cluster team; Ana Riesgo for sharing demosponge phylogenies; the reviewers, for their suggestions, in particular rev. 1; and Juan Daniel Montenegro Cabrera for suggesting the term "morpholog".

### Peer review information

### Review history

The review history is available as Additional file 2.

### Authors' contributions

NP and FR conceived the project. NP, FR, JMM, MM, and MS designed the project. NP and FR performed the analysis. MS and MM performed additional sequence-based analysis. NP, FR, JMM, and DA wrote the manuscript. All authors read and approved the final manuscript.

### Funding

 NP was funded by MSCA individual fellowship 101031984/DeCoDe Platy. FR has received funding from the European Union's Horizon 2020 research and innovation program under the Marie Sklosowska-Curie grant agreement No. 764840/IGNITE as well as EMBL International PhD Program. MS acknowledges support from the National Research Foundation of Korea (NRF), grants [2020M3-A9G7-103933, 2021-R1C1-C102065, 2021-M3A9-I4021220], Samsung DS research fund and the Creative-Pioneering Researchers Program through Seoul National University. The work was supported by the European Research Council Advanced grant 788921/NeuralCellTypeEvo (DA).

### Availability of data and materials

The dataset supporting the conclusions of this article is available in the Zenodo repository [108]. The predicted *Spongilla* protein structures are deposited in ModelArchive [109].

The code that produced the analysis as well as the figures, including a notebook with instructions on how to obtain the data, is available online. In particular:

• Project name: MorF (Morpholog Finder)
• Project home page: https://git.embl.de/grp-arendt/MorF/
• Archived version: v1.2, available on Zenodo [110] and GitLab [111].
• Operating system(s): set-up depends on a UNIX environment; analysis is platform-independent
• Programming language: bash, Python, Perl
• Other requirements: for an exactly reproduced environment refer to scripts, conda environment description.
• License: GPL 3.0

## Declarations

### Ethics approval and consent to participate

Not applicable.

### Consent for publication

The publication contains no personal data in any form.

### Competing interests

The authors declare that they have no competing interests.

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

## 

