## [**Additional file 2.** Review history. · Genome Biology]

Review History

First round of review

Reviewer 1

Were you able to assess all statistics in the manuscript, including the appropriateness of statistical tests used? There are no statistics in the manuscript.

Comments to author:

Review of the manuscript „Beyond sequence similarity: cross-phyla protein annotation by structural prediction and alignment" submitted by Ruperti et al.

In the present manuscript, the authors propose a workflow, CoFFE, that exploits protein structure prediction in combination with a subsequent protein-structure based database search to transfer functional annotation between proteins over evolutionary distances that become challenging for sequence similarity-based approaches. Using the sponge *Spongilla lacustris* as a use case, they initially benchmark CoFFE via a comparison to EggNOG-mapper. Subsequently, they apply CoFFE for the functional annotation of the *S. lacustris* proteome shedding further light on signaling pathways and ROS metabolism and redox control. Eventually, they report evidences for horizontal gene transfer having shaped the set of genes that characterize mesocytes. Overall, I consider this approach timely, valid and interesting. However, I see a number of issues that should be taken care of, which mostly deal with the method itself and with the acquisition of mesocyte marker genes. Please note that Figs. 1 - 4 in the main manuscript are referred to as Figs. 7 - 10 both in the text and in the figure legends. I will use the intended numbering, i.e. starting with Fig. 1.

Major issues

1. The main goal of the CoFFE pipeline is the annotation transfer between proteins over large evolutionary distances using the similarity of their predicted 3D structure as evidence. Evolutionary relationships between sequences, which are thus far considered as the backbone of an annotation transfer, are only considered indirectly, if at all. The most relevant question that should be addressed via a benchmark is therefore 'What is the precision of CoFFE in identifying functionally equivalent proteins, and, in turn, how much functional deviation is possible until the structures become dissimilar to an extent that they no longer generate a significant hit?'. I fear that the benchmark thus far does not provide a relevant answer to these questions.

a. The comparison is restricted to the subset of proteins where the EggNOG-Mapper provides a functional annotation. In these cases, the sequence similarity is conserved enough to warrant both an orthology assignment and a functional annotation transfer. These instances should be the trivial cases for CoFFE. I think it is essential that the authors show that beyond these 'simple' cases, the CoFFE approach maintains its precision. This could be done, for example, by comparing the functionally annotated protein sets in distantly related model organisms, e.g. yeast - human or Arabidopsis - human, and assessing how often a CoFFE query-hit pair represents proteins that are not significantly similar, and which mutually lack orthologs in the other species, share the same function.

b. The authors argue with Table 3 that the proteins with the highest structural similarity are in most cases also orthologs. I see two issues here: First, CoFFE is not concerned with the identification of orthologs, but with the identification of functionally equivalent proteins. Hence, it is not entirely clear how the information from Table 3 integrates into the line of argumentation. The ortholog conjecture could be used as the connecting element, however, it is clear that there are many examples where orthologs have indeed been functionally diverged, especially when evolutionary distances become large. Second, Table 3 provides only the information about whether or not the best non-species hit is an ortholog, it does not specify which species contributes the best hit. In the worst case, the best non-species hit for a human protein is a chimpanzee protein. And that this identifies in more the 95% of the cases the ortholog is trivial. If the focus is indeed on long evolutionary distances, it would be interesting to learn in how many cases the best hit from another phylum/kingdom/domain is a best hit.

2. CoFFE applies a unidirectional search to identify proteins with a similar predicted structure. Beyond what threshold are hits considered to provide evidence about the function of the query protein, and how is this threshold justified? Along the same lines, what is the variation of functions represented by the top N hits within a certain score margin? In how many cases do the best and the second best hit result in marginally different similarity scores but are annotated with different functions?

3. The findings about the horizontal acquisition of mesocyte marker genes are interesting, and the authors provide several lines of evidence to convince the reader that these observations are not due to contaminations in the sequence data. Still, the authors should provide additional information in support of the HGT hypothesis, since there have been a plethora of spurious reports about HGT in Eukaryotes over the past years:

a. The claim that the candidate genes are flanked by metazoan genes in an assembly is not backed up by data. To rule out assembly artefacts, it would be relevant to see that individual (long) reads cover both the HGT candidate and the flanking metazoan genes

b. The adaptation of the GC content and of the codon usage is a process that requires time. It would be helpful if the authors could comment on how much time the candidates had to adjust to the genomic landscape of the host genome. In this context, it would be nice to see the placement of *S. lacustris* in the tree shown in Fig. 4

c. More precise information is missing about the likely source taxon as well as about the prevalence of the gene in the phylogenetic clade the source taxon is embedded into. In fact, can the authors rule out that the transfer occurred in the opposite direction, i.e. the sponge is the source species? Likewise, information about the extent of sequence similarity between the xenologs in source and host taxon would be helpful.

Minor issues

1. I appreciate the acronym CoFFE, but I have difficulties with it for two reasons: First, the 'E' connects to EggNOG, but I do not see from the outline of the algorithm on page 2 where EggNOG is used. In the same line, Fig. S1 shows a compulsory connection between the outcome of the FoldSeek results and a downstream EggNOG-Mapper. However, again from the description on page 2, this is not obvious. This should be clarified. Second, CoFFE is very similar to COFFEE, a consistency based objective function for alignment evaluation. Since both CoFFE and COFFEE might be used by the same community, it might be worthwhile to

consider a different acronym. This should be taken only as a hint and not a request for a change

2. The authors do make no comments about the assignments made by EggNOG-mapper but that are not reproduced by CoFFE. The community is certainly interested in understanding the relevance of this finding.

3. I do not see how the sharing of a Pfam domain between the CoFFE hit and the EggNOG-Mapper hit can be taken as evidence that the two predictions agree

4. Fig 1A - The authors state that the *S. lacustris* structure prediction results in similar average pLDDT scores than other model organisms. However, the median seems quite a bit lower, and superficially one could claim that *S. lacustris* and *D. discoideum* give the lowest scores. Has the NULL hypothesis of equal score distributions been tested?

5. The authors state that with their analysis *S. lacustris* has now a functional annotation level that is comparable to *C. elegans*, a widely used model organism. And they refer to Fig. 1B. I wonder whether this is fair to state: First, no CoFFE analysis has been performed with the *C. elegans* proteome. This would likely increase the fraction of annotated proteins. Second, I trust that the functions of many *C. elegans* proteins have experimentally verified, whereas this is not the case for the sponge

6. I am a bit confused about the outcome of the comparison between CoFFE and EggNOG (Fig. 1C). In about a third of the cases the hits belong to the same gene family. What exactly does this mean given that both approaches aim at providing a functional annotation

7. In Figure 3, the authors present the extension of the ROS metabolism / control based on the CoFFE analysis. I checked only two proteins, FLC3 and FRRS1. FLC3 seems to be a fungal protein, according to orthology databases. It might be that case that its phylogenetic distribution is underestimated due to a rapidly diverging sequence, and it is also present in animals. In the case of FRRS1, however, the situation is different. According to EggNOG, orthologs for this protein are present throughout the eukaryotes http://eggnog5.embl.de/#/app/results?seqid=Q8K385&target_nogs=KOG4293#KOG4293_datamenu. This implies that this gene would have been found also with conventional sequence similarity-based searches. Is this correct? What fraction of the genes marked in red in Fig. 3 are only identified by CoFFE?

8. I suggest to replace the term 'phylogenetic context' with 'phylogenetic profile'

9. Instead of using BlastP, it is probably better to search for distantly related / very dissimilar proteins in nrProt using more sensitive approaches, such as Psi-Blast

Reviewer 2

Were you able to assess all statistics in the manuscript, including the appropriateness of statistical tests used? Yes, and I have assessed the statistics in my report.

Comments to author:

This paper is on the vanguard of the exciting ability to use structural similarity rather than sequence similarity to evaluate protein homology. There are likely many cases where homology is no longer apparent at the sequence level but is readily evident at the morphological level. Tools that can identify and evaluate this structural homology will be very important as more high quality genomes are produced for organisms that we otherwise know very little about.

The paper presents an approach for integrating several tools, including some that are very new, to achieve these objectives. They convincingly show that this is highly illuminating in the context of annotating the *Spongilla* protein set. This work shows that these methods are congruent with other approaches, where this can be examined, and convincingly provide new biological insight.

My primary concern is that the paper presents this contribution as a pipeline named CoFFE. A pipeline is an implementation of a method (a tool) that can be evaluated on its own and extended to other datasets. But no software is presented. There is nowhere to download and learn to use the pipeline (or I missed it). This makes it difficult to review the methods, and in my mind falls short of the expectations set out in the abstract. The authors should recontextualize their contribution as a method used in this particular analysis of a sponge proteome, rather than an implementation. Or (preferably) make the software available according to standard practices (eg a github repo where others can download the tool to use on their own dataset, with documentation and worked examples) that would be consistent with the presentation of a new "pipeline".

I am excited to use this approach, and hopefully this implementation, in my own work. I am sure others will be too.

Specific comments:

page 1 line 54: "protein similarity" -> "protein sequence similarity"

page 5 line 45: "strengthen the case for" reword. This is a hypothesis you are testing, current wording makes it sound like you already decide this was the case.

page 6 line 1: this is a weak case for HGT without assessing presence in sister groups to animals.

Reviewer 3

Were you able to assess all statistics in the manuscript, including the appropriateness of statistical tests used? There are no statistics in the manuscript.

Comments to author:

In this manuscript, Ruperti et al combine recent progress in protein structure prediction (AlphaFold) with progress in structure comparison (Foldseek) to improve the prediction of

gene function in the sponge *Spongilla lacustris*. They then combine these new annotations with single-cell RNA-seq data to gain insight into cellular processes in this sponge.

The method and results presented are of wide interest to make sense of the genomics of non model species, which is increasingly important to understand both biodiversity and functional genomics.

It was a pleasure to review this manuscript. While I have some questions and comments, it is overall very well written, and presents an exciting approach and interesting results. I commend the authors on the care with which the methods were not only executed, but also presented clearly and in detail.

Major comments:

1- All the genome-wide interpretation of new annotations would be much improved by providing information on the level of detail of the annotations. For example CAFA (<https://doi.org/10.1186/s13059-019-1835-8>) uses depth in the Gene Ontology graph. Otherwise it is difficult to compare e.g. the % of genes annotated between nematode and this sponge. If most of the annotations here are "binding" molecular function, it is not informative. The examples which are detailed indicate that annotations can be quite informative, but on the other hand for 1/3 of genes with a known ortholog CoFFE only finds the correct family.

2- The comparison to EggNOG is interesting, but limited by the capacities of that tool.

2a- On the one hand, it is clear that a less specific but more sensitive approach would annotate more genes. Thus it would be interesting to compare CoFFE to InterProScan, which like CoFFE has the potential to provide sensitive if not specific annotations.

2b- On the other hand, since there were many duplications in bilaterian animals since the divergence with sponges, in many cases there will not be one correct ortholog assignment, and EggNOG might be over-classifying. In which case finding a different ortholog might not be a mistake. Thus it would be interesting to either compare to a tool which avoids such over-classification, or to provide some measure of how often such groups of co-orthologs occur. I do not have a specific solution to offer here but at least a discussion of the issue would be welcome.

3- For the benchmark of CoFFE on model organisms to be informative relative to the aim of annotating species such as sponge which are distant from model organisms, I would advise to remove from the query database not only the model organisms but their whole lineage (e.g. vertebrates or eudicots).

Minor comments:

"*Spongilla lacustris*, an early-branching animal": What does early-branching mean here? I think the main point in context is that has diverged very long ago from any well annotated model organism.

I don't understand what the following sentence means: "Considering that careful evaluation of sequence-based phylogenetic trees can identify a protein's evolutionary history [55], it will be interesting to explore the extent to which annotations based on structure provide better predictors of function than evolutionary homology."

"Relative alignment length (FS) (percentage of the query (Spongilla) structure aligned with the best target structure) does not correlate too strongly": Do you mean "correlates weakly"?

I found ref 8 was now published at <https://www.nature.com/articles/s41592-022-01488-1>

Availability of data and materials: the link to zenodo is incorrect.

Response to Referees: Ruperti, Papadopoulos et al.

Introduction

We thank the reviewers for their kind comments and constructive feedback. We made a number of major changes that address the reviewers' concerns and significantly improve the manuscript.

- We recognized the necessity for increased clarity in the manuscript. We therefore introduce the term “morpholog” to describe groups of proteins with similar structures. Accordingly, we renamed the pipeline from “CoFFE” to “MorF” (**M**orpholog **F**inder).
- In the Results section, we revisited the comparison of MorF to sequence-based annotations to acknowledge the usage of the ortholog conjecture and discuss the role of EggNOG/EggNOG-mapper in our pipeline. In a new subsection, we show that proteins with similar structures overlap significantly in functional annotation (GO terms and EC numbers) across long evolutionary distances, and even when homology is not detectable by conventional, sequence-based methods.
- We added additional analyses such as comparisons of MorF annotations to more sensitive HMMER-based searches and extended sequence based searches for *Spongilla* cell type marker genes. All analyses are summarized in the main text and expanded in additional supplemental notes, figures or tables.
- We reworded and weakened the paragraph claiming HGT origin of mesocyte marker genes.

The direct answers to the reviewers' comments and suggestions follow. Text passages that are copied verbatim from the manuscript are highlighted in blue.

Reviewer 1

Major issue 1

The main goal of the CoFFE pipeline is the annotation transfer between proteins over large evolutionary distances using the similarity of their predicted 3D structure as evidence. Evolutionary relationships between sequences, which are thus far considered as the backbone of an annotation transfer, are only considered indirectly, if at all. The most relevant question that should be addressed via a benchmark is therefore 'What is the precision of CoFFE in identifying functionally equivalent proteins, and, in turn, how much functional deviation is possible until the structures become dissimilar to an extent that they no longer generate a significant hit?'

1a

The comparison is restricted to the subset of proteins where the EggNOG-Mapper provides a functional annotation. In these cases, the sequence similarity is conserved enough to warrant both an orthology assignment and a functional annotation transfer. These instances should be the trivial cases for CoFFE. I think it is essential that the authors show that beyond these 'simple' cases, the CoFFE approach maintains its precision. This could be done, for example, by comparing the functionally annotated protein sets in distantly related model organisms, e.g. yeast - human or Arabidopsis - human, and assessing how often a CoFFE query-hit pair represents proteins that are not significantly similar, and which mutually lack orthologs in the other species, share the same function.

We compared functional descriptions of morphologs between *S. cerevisiae*/*A. thaliana* and *H. sapiens* for which no evidence of homology existed, as suggested by the reviewer. For comparative purposes, we focused on enzymes. An enzyme's EC number is a four-digit numerical description of its function, with each number representing a progressively finer classification of the enzyme. The higher the EC number overlap between two proteins the more similar their function; if two different enzymes catalyze the same reaction they will have the same EC number. For instance tripeptide aminopeptidases have the code "EC 3.4.11.4", with each component indicating a following group of enzymes:

- EC 3: hydrolases
- EC 3.4: hydrolases that act on peptide bonds
- EC 3.4.11: hydrolases that cleave the amino-terminal amino acid from a polypeptide
- EC 3.4.11.4: hydrolases that cleave the amino-terminal amino acid from a tripeptide

The results of this analysis are described in Suppl. Note. H and summarized in the main text in the Results section, subsection "Morphologs share function over long evolutionary distances":

"As a next step, we sought to explore whether MorF can be used for functional annotation in cases where the evolutionary distance is too large for sequence-based approaches. To test this, we performed Foldseek searches for the predicted *S. cerevisiae* and *A. thaliana* proteomes against *H. sapiens*, and identified morphologs that lacked evidence of homology based on sequence. It is important to note that it is impossible to know for sure whether or not these represent homologs or protein structures that evolved via convergent similarity.

Nevertheless, for the remaining candidates, we tested their functional similarity by examining the overlap of their Enzyme Commission (EC) number where available. The EC number is a four-digit numerical description of enzyme function, with each number representing a progressively finer classification of the enzyme. Agreement on the first digit indicates two proteins are in the same broad enzyme class (oxidoreductases, hydrolases, ligases, etc.), while complete agreement means that they catalyze the same reaction.

For yeast, 109/145 (75%) enzymes agreed with their human morphologs on three of four EC positions and 53/145 (36.5%) agreed on all four. Similarly, for Arabidopsis, 357/532 eligible enzymes agreed to the third EC position (67%) and 176/532 (33%) had identical EC numbers. These results indicate MorF can accurately predict function even in cases where protein homology is unclear due to large evolutionary distances. Furthermore, this is consistent with other work in the field that has demonstrated that structure similarity uncovers homologs between *Homo sapiens* and different *Saccharomyces* species [21] using a similar methodology. We eagerly expect more insights on this topic in the coming months and years."

The accompanying code is available online¹, and we have amended the manuscript to include these new results.

1b

The authors argue with Table 3 that the proteins with the highest structural similarity are in most cases also orthologs. I see two issues here: First, CoFFE is not concerned with the identification of orthologs, but with the identification of functionally equivalent proteins. Hence, it is not entirely clear how the information from Table 3 integrates into the line of argumentation. The ortholog conjecture could be used

¹ https://git.embl.de/grp-arendt/CoFFE/-/blob/main/analysis/revision-remote_species.ipynb

as the connecting element, however, it is clear that there are many examples where orthologs have indeed been functionally diverged, especially when evolutionary distances become large.

We appreciate the reviewer's comments and have clarified the text to better discuss the goal of our pipeline. Importantly, our goal is not to replace homology-based annotations or suggest structure is a better predictor of function than homology. Rather, our goal is to extend the toolkit for identifying homologs separated by large evolutionary distances. As we show, MorF annotations are highly consistent with sequence-based approaches for identifying more closely-related homologs. In a majority of cases, both MorF and sequence-based methods identify orthologous proteins, and in most of the rest identify homologous members of the same family. These may either correspond to paralogs or to incorrectly classified orthologs, which is a substantial problem for sequence-based orthology algorithms in fast-evolving proteins or at large evolutionary distances (see Natsidis et al. 2021). Regardless, both orthologs and paralogs often share highly similar functional annotations, which we also confirm by comparing MorF and eggNOG annotations of GO terms, EC numbers, and semantic similarity. Again, we would argue this is not surprising, as most homologs share broadly similar functions, even if they are not strictly orthologs (e.g. paralogous G-protein coupled receptors are nearly always membrane-bound receptors that activate G proteins upon binding of a ligand).

In a very few cases, MorF and sequence-based approaches identify apparently non-homologous proteins. It is tempting to speculate that structural similarity may be a better guide to function than homology, yet this must be experimentally validated, which is outside the scope of this manuscript.

To better clarify the goals of our pipeline we have altered the text in several places.

In the beginning of the Background section:

"Knowledge of protein function is crucial for interpreting many types of high-throughput molecular datasets. Since protein functional studies are limited to a few model species, amino acid sequence similarity has been used to infer the function of protein homologs [1]. However, homology detection over longer evolutionary distances remains challenging owing to the decay of protein sequence similarity that abolishes evidence of historical continuity. This presents a severe bottleneck for inferring protein function across a wide expanse of the tree of life, particularly in distant organisms where many proteins fall in the "twilight zone", only sharing a sequence identity between 10 – 20% with proteins in characterized models [2, 3].

A way to venture more deeply into the twilight zone is to **use structural similarity for homology detection, as structure is more conserved in evolution** [4]. Until recently, this was not feasible since predicting protein structures from amino acid sequence required prior inference of sequence homology [5]. This has changed with the advent of AlphaFold [6], a deep learning AI system that can predict de novo protein structures with atomic resolution, together with novel approaches for identifying similar structures in large databases [7]. Protein structures can now be predicted by AlphaFold for entire proteomes, and then aligned to structures from model systems with characterized functions."

In the beginning of the Results subsection "MorF annotations agree with sequence-based annotation transfer":

"Traditional approaches for functional annotation use protein sequences to identify orthologous or homologous proteins. These methods exploit the fact that homology is often the best predictor of shared function, and effectively treat functional annotation synonymously with assessment of

homology, although it is known that divergence of function within orthologs is possible. As a next step, we evaluated whether MorF annotations are congruent with those produced via traditional sequence-based homology approaches.”

Furthermore, we compared EC numbers between morphologs with no detectable homology and found that enzyme function often agrees with structural similarity (as described above). Finally, we also performed comparisons of GO term overlap, semantic similarity and depth between sequence and MorF-based annotations of *Spongilla* proteins following CAFA [1] protocol, as suggested by reviewer 3. This analysis has been added as Supplemental Note E and summarized in the main text:

“We also examined consistency between MorF and sequence-based annotations for *Spongilla* by comparing GO term overlap, semantic similarity and depth [22], drawing on the annotation comparisons of the CAFA challenge [23]. Almost all (99.9%) proteins annotated by both strategies have at least some overlap in GO terms with 60.6% being identical and 39.3% partially overlapping to various degrees (Suppl. Fig. 6A). GO term semantic similarities between proteins with overlapping (but non-identical) GO annotations additionally reach an average score of 80–90% in the molecular function ontology (Suppl. Fig. 6B).”

With these additional analyses we are further convinced that structural similarity is able to recover functional annotations beyond the ortholog conjecture.

Second, Table 3 provides only the information about whether or not the best non-species hit is an ortholog, it does not specify which species contributes the best hit. In the worst case, the best non-species hit for a human protein is a chimpanzee protein. And that this identifies in more than 95% of the cases the ortholog is trivial. If the focus is indeed on long evolutionary distances, it would be interesting to learn in how many cases the best hit from another phylum/kingdom/domain is a best hit.

We thank the reviewer for this suggestion, and Reviewer 3 (issue 3) for raising a similar issue. As the reviewer notes, it is likely that the best non-species hit for any query organism will be from the closest related species in our search databases (except where horizontal gene transfer has occurred).

We revisited Table 3 to exclude species that belong in the same taxonomic group when evaluating the top morphologs. We reanalyzed our comparison of MorF and sequence-based methods in model species by excluding their close relatives, such that the best MorF hit will necessarily be a protein in another animal phylum. Again, we found that the large majority of MorF and sequence-based best hits are homologs. This suggests that structural similarity is indeed a viable method for homology detection even over longer evolutionary distances. We updated the manuscript to reflect this change.

In Results, subsection “MorF annotations agree with sequence-based annotation transfer”:

“We first examined the agreement between morphologs and homologs on available predicted structures of model organisms. We aligned AlphaFoldDB against itself and kept for each query the top morpholog outside the species taxonomic unit. This ensured that MorF would not be identifying quasi identical proteins from closely related species (e.g. *M. musculus* and *R. norvegicus*), simulating a realistic use case where MorF would be used to annotate a non-model organism without well-studied close relatives (Suppl. Table 3). We assessed performance by comparing the eukaryotic orthologous group of the top morpholog to that of the query protein, as defined by the EggNOG v5.0 database. MorF routinely identifies 75-90% of all available homologs, indicating that it can successfully reproduce sequence-based homology inference.”

In Appendix H: Structure-sequence agreement in model species, we updated Table 3:

species	#queries with out-of-group targets	#eligible queries	%eligible agreement	taxonomic group
H. sapiens	20,392	13,838	74.30%	Vertebrata
M. musculus	21,507	15,084	75.05%	Vertebrata
R. norvegicus	19,246	13,180	74.60%	Vertebrata
D. rerio	24,626	13,303	75.36%	Vertebrata
D. melanogaster	13,418	8,828	79.69%	Arthropoda
C. elegans	19,599	11,177	67.29%	Nematoda
S. mansoni	13,845	1,730	91.85%	Nematoda
A. thaliana	27,357	19,800	83.78%	Eudicots
G. max	55,759	32,239	85.40%	Eudicots
Z. mays	38,818	18,644	85.47%	Monocots
O. sativa subsp. japonica	41,794	20,306	86.51%	Monocots
S. cerevisiae (str. ATCC 204508/S288c)	6,013	4,347	83.41%	Saccharomycetales
S. pombe (str. 972/ ATCC 24843)	5,104	3,982	90.51%	Saccharomycetales
C. albicans (str. SC5314/ ATCC MYA-2876)	5,973	4,118	84.41%	Saccharomycetales
C. carrionii	11,169	7,040	81.29%	Chaetothyriales
P. lutzii (str. ATCC MYA-826/Pb01)	8,791	5,086	84.19%	Eurotiomycetes
S. schenckii (str. ATCC 58,251/ de Perez 2211183)	8,652	6,349	78.41%	Sordariomycetes
T. brucei brucei (str. 927/4 GUTat10.1)	8,476	4,303	60.26%	Trypanosomatida
T. cruzi (str. CL Brener)	19,026	8,782	56.56%	Trypanosomatida
L. infantum	7,914	4,423	56.14%	Trypanosomatida
D. discoideum	12,584	6,312	63.96%	Amoebozoa

It now lists the broad taxonomic group to which we assigned each species. These reflect clades, orders, phyla, or even kingdoms, depending on the prevalence of related organisms in the database. We updated the accompanying text as well:

“As proof of principle we explored the power of structural similarity to identify orthologous genes for a number of model species whose proteomes already had predicted structures in AlphaFoldDB. We aligned AlphaFoldDB against itself using Foldseek. For each query protein we kept the best target from outside the taxonomic group and compared the eukaryotic orthogroup assignment. The best structural hit overwhelmingly belonged to the same orthogroup as the query (Suppl. Table 3).

We used the Proteins API [112] to obtain the species name for each entry [113]. We decorated each entry with its orthogroup assignment from the EggNOG database [17] (v5.0). Not all proteins currently in AlphaFoldDB could be assigned an orthogroup; we ignored comparisons where the query or its best target were missing the orthogroup assignment. The full table can be found in the corresponding note- book [114].

We compared the remaining eligible cases and found overwhelming agreement between the eukaryotic orthogroup of the best structural target and the orthogroup of the query protein, confirming that structural similarity can, in principle, detect homology relationships. Though the available targets were restricted to be outside the clade, order, phylum, or even kingdom, top morphologs still are homologs in the large majority of cases, a very encouraging indication that MorF is able to detect homology over long evolutionary distances.”

Spongilla lacustris is only distantly related to animal models and is as such the intended use case for our pipeline. For *Spongilla* proteins the best hit is always a protein from a different animal phyla. We show MorF consistently identifies cross-phyla homologs in ~90% of cases, further confirming what Table 3 suggests, namely that structural similarity is well-suited for homology detection across long evolutionary distances.

Major issue 2

CoFFE applies a unidirectional search to identify proteins with a similar predicted structure. Beyond what threshold are hits considered to provide evidence about the function of the query protein, and how is this threshold justified?

The goal of assigning a threshold to Foldseek searches is to ensure best hits to represent good structural matches. Only then, we can confidently use the morphologs for functional transfer. We chose our threshold based on two factors. First, personal communication with the Foldseek authors suggested that hits with bit scores > 100 confidently identify functionally similar proteins. Second, plotting Foldseek bit scores against frequency of search results, we identified a clear bimodal distribution with a local minimum at a bit score of about 150 (e^5) (see attached histogram). Therefore

we decided to err on the side of caution and transfer the function of structures with best hit log bit scores > 5 (≈ 150). In the methods section we describe the reasoning behind this threshold:

“For each search we kept the Foldseek hit with the highest corrected bit score in each database and aggregated the three result tables (AlphaFoldDB, PDB, SwissProt) into one. We imposed a bit score cutoff of e^5 on Foldseek hits based on their bimodal distribution and personal communication with the Foldseek authors.”

Along the same lines, what is the variation of functions represented by the top N hits within a certain score margin? In how many cases do the best and the second best hit result in marginally different similarity scores but are annotated with different functions?

We defined “top N hits” as the morphologs in the 90th percentile of the bit score range for each query. For each protein we assessed the functional overlap of the top N morphologs with the best morpholog. We quantified this as the average agreement of the EC numbers of the morphologs with

the EC number of the top morpholog (also see Major Issue 1). We observed an average EC overlap of 3.7 - close to complete agreement, with only 10.88% of cases in which the first and second best morphologs do not have identical EC numbers.

This analysis was summarized in the Results section, subsection “Morphologs share function over long evolutionary distances”:

“We also assessed the consistency of the functional annotation for top morphologs in *Spongilla*. For each protein we queried the EC number of all morphologs in the 90th percentile of the Foldseek score range. This serves as an indirect way of validating that significant structural similarity correlates with functional conservation. In the 7072 cases that we could evaluate, the top morphologs were close to identical to the EC number of the best morpholog (average agreement 3.7 positions; see Suppl. Note D).”

For more details please refer to Suppl. Note D (Functional similarity of top morphologs).

Major issue 3

The findings about the horizontal acquisition of mesocyte marker genes are interesting, and the authors provide several lines of evidence to convince the reader that these observations are not due to contaminations in the sequence data. Still, the authors should provide additional information in support of the HGT hypothesis, since there have been a plethora of spurious reports about HGT in Eukaryotes over the past years.

3a

The claim that the candidate genes are flanked by metazoan genes in an assembly is not backed up by data. To rule out assembly artefacts, it would be relevant to see that individual (long) reads cover both the HGT candidate and the flanking metazoan genes.

We thank the reviewer for this useful suggestion. We currently only have a preliminary assembly for the *Spongilla* genome using PacBio HiFi reads. Following the reviewer’s suggestion we looked for reads containing genes from putative HGT events and genes of metazoan origin. However, we only found very few cases where reads contained multiple genes at all, mostly due to very long intergenic regions and repeat content. Thus, at the current state of analysis we were unable to validate HGT events using our current data. Although we believe this is a viable approach, it will need either additional sequence data or longer reads (e.g. nanopore). We still believe the evidence warrants the interpretation that these genes originated via HGT, but now discuss alternative interpretations and feel obliged to weaken the overall claim for HGTs while at the same time stressing MorFs ability to detect genes of non-Metazoan origin.

The amended text can be found in the Results subsection “Polysaccharide hydrolysis in enigmatic mesocytes”:

“Although our analyses suggest these genes may have originated via HGT [64], it is important to consider alternative explanations. One possibility is that these genes are the results of widespread bacterial contamination in sponges genomes and proteomes. Other possibilities include the repeated loss of these genes in other animal lineages, or the convergent evolution of similarity with bacterial proteins. Additional confirmation of their presence in sponge genomes, or evidence of RNA transcripts in sponge cell nuclei would help validate the hypothesis they arose via HGT. Regardless of

the source of the genes, MorF annotations provided a novel hypothesis for the elusive function of sponge mesocytes, helping uncover new aspects of sponge biology.”

3b

The adaptation of the GC content and of the codon usage is a process that requires time. It would be helpful if the authors could comment on how much time the candidates had to adjust to the genomic landscape of the host genome. In this context, it would be nice to see the placement of S. lacustris in the tree shown in Fig. 4.

The information regarding potential HGT event timing has been included in the updated version of the manuscript as well as Table 6. Likewise, the phylogenetic position of *S. lacustris* has been added to Fig. 4.

“The prospect of HGT is tantalising. Proteins with enzymatic functions like the ones in the *Spongilla* candidates (polysaccharide hydrolases and metallopeptidases) have been proposed to be horizontally transferred in *A. queenslandica*, a marine demosponge, *S. rosetta*, a choanoflagellate, and *M. leidy*, a ctenophore [61, 62, 63]. Furthermore, *Spongilla* genes c97022_g1 and c103983_g1, a putative aminohydrolase and metallopeptidase respectively, are not only broadly distributed within sponges, but can also be found in the proteomes of choanoflagellates *S. rosetta* and *M. brevicollis* (Suppl. Note L). Furthermore, the *S. rosetta* targets with highest similarity to the *Spongilla* sequences had already been identified as horizontally transferred genes [62]. This would tentatively place this HGT event at least before the split of choanoflagellates and animals (more than ~500mya). Similarly, the phylogenetic distribution of c102757_g1, c95037_g1, and c102838_g2 (yocJ, sleB, Chitinase class I) would indicate that it was acquired with the colonisation of freshwater environments (~15-300mya).”

3c

More precise information is missing about the likely source taxon as well as about the prevalence of the gene in the phylogenetic clade the source taxon is embedded into.

In order to narrow down potential source taxa of HGT candidates we performed blastp searches against non-redundant (nr) protein sequence and metagenomic (env_nr) databases as well as EggNOG 5.0 sequence searches. The phylogenetic distribution of blastp results as well as EggNOG searches in some cases allowed narrowing down potential source taxa such as Alphaproteobacteria or CFB-group Bacteria. Results of this analysis can be found in the updated Table 1 and the new Table 6.

In fact, can the authors rule out that the transfer occurred in the opposite direction, i.e. the sponge is the source species?

Although we cannot categorically rule this possibility out, we consider this far less likely owing to the widespread occurrence of the proteins in bacteria compared to their highly restricted distribution in animals. Furthermore, some of the potential host taxa are known freshwater symbionts, e.g. Planctomycetes (Kohn *et al.* 2020).

Likewise, information about the extent of sequence similarity between the xenologs in source and host taxon would be helpful.

This information has been added in Table 6. The xenologs share between 40 and 60 % sequence identity between source and host taxon.

Minor issue 1

I appreciate the acronym CoFFE, but I have difficulties with it for two reasons: First, the 'E' connects to EggNOG, but I do not see from the outline of the algorithm on page 2 where EggNOG is used. In the same line, Fig. S1 shows a compulsory connection between the outcome of the FoldSeek results and a downstream EggNOG-Mapper. However, again from the description on page 2, this is not obvious. This should be clarified.

We thank the reviewer for this remark and have now updated the manuscript to explicitly discuss how EggNOG-mapper is used in the pipeline. Although EggNOG searches are not mandatory for functional annotation transfer, they are very convenient as they enable straightforward comparisons between MorF and emapper results:

*“After removing lower-quality matches, we retrieved functional annotations for the best morphologs using EggNOG-mapper (emapper) [16], a state-of-the-art orthology database [17], and then assigned these annotations to the protein in *Spongilla*. This produced annotations for slightly more than 60% of the proteome (25,232 proteins), representing an increase of approximately 50% compared to when *Spongilla* protein sequences were directly searched with emapper. Whereas the usage of emapper is not compulsory for MorF, it provided functional descriptors like EC numbers or GO terms for orthologous groups, facilitating later programmatic comparisons to sequence based methods. However, for downstream biological analysis, gene names and descriptions remain the most succinct, human-readable proxies for protein function. We therefore decided to use the preferred name and description of the best morpholog for each *Spongilla* protein assigned by emapper.”*

Second, CoFFE is very similar to COFFEE, a consistency based objective function for alignment evaluation. Since both CoFFE and COFFEE might be used by the same community, it might be worthwhile to consider a different acronym. This should be taken only as a hint and not a request for a change.

We changed the name to 'MorF' (Morpholog Finder) after introducing the term 'morpholog'. This offers increased clarity and avoids collisions with COFFEE/T-COFFEE. Please also refer to the extended answer to reviewer 2 who shares similar justified concerns about the acronym.

Minor issue 2

The authors make no comments about the assignments made by EggNOG-mapper but that are not reproduced by CoFFE. The community is certainly interested in understanding the relevance of this finding.

There are two different groups of proteins that fall into this category:

1. *Proteins that were only annotated by EggNOG-mapper and didn't receive a MorF annotation. These cases were mentioned in Suppl. Note B and shown in Suppl. Fig. 3 (described as 'No structure / Sequence') but have not been discussed in detail yet. Therefore we added the following discussion and possible explanation to Suppl. Note B:*

“Proteins with only sequence based annotation (“No structure / Sequence”) have on average relatively high pLDDT values (Suppl. Fig. 3A). Nevertheless, their Foldseek bit score is too low (Suppl. Fig. 3D) to pass the bit score threshold for annotation. One possible explanation is that this category entails relatively short proteins (Suppl. Fig. 4A), which are shown to be suboptimal for structural searches, as they often do not pass (statistical) thresholds.”

2. *Proteins that were annotated differently by EggNOG-mapper and MorF (equivalent to 'no agreement' in Fig. 1C).*

These cases have not been discussed in the manuscript so far. We performed additional analyses and present the results in a new supplementary note Suppl. Note C) and figure (Suppl. Fig. 5):

"[...] MSA sizes and corrected bit scores (Fig. S5E-F) are lower for proteins in the 'no agreement' category compared to proteins with (partially) overlapping sequence and structure annotations. At the same time, the structure prediction quality for proteins (pLDDT) in the 'no agreement' category are only marginally lower (Fig. S5C). Comparing Foldseek corrected bit scores across the annotation agreement categories broken down to multiple pLDDT buckets (Fig. S5B) shows that even even in the cases of high prediction quality (pLDDT = 90 - 95%) the corrected Foldseek bit scores for the 'no agreement' cases are very low, potentially leading to uncertainty in the structurally guided functional annotation. There are two potential reasons for this observation: First, Foldseek search results depend on the content of the structural databases that Foldseek searches against. Even very good structural predictions of sponge proteins can have low Foldseek bit scores if there are no morphologs present in the Foldseek database. With the ever expanding prediction of structures in the AlphaFold database, this issue will possibly be mitigated in the future. In fact, during the review process of this manuscript, a new version of AlphaFoldDB has been released, containing protein structures from all UniProt entries. Second, comparing query lengths of proteins in different annotation agreement levels (Fig. S5D), we observed a large proportion of long proteins in the 'no agreement' category. Longer proteins often are composed of multiple domains that are linked via flexible linkers. Manual inspection showed that ColabFold correctly predicts the (globular) domains, leading to overall high pLDDT values, while predicting the flexible linker seemingly in a random fashion. This in turn leads to a random positioning of the domains relative to each other. This poses an issue for the subsequent Foldseek search in which only one of the domains can be superposed correctly, leading to the observed low corrected bit score. However, the extent to which this leads to incorrect functional annotation is still unknown and is outside the scope of the study.

Taking the possible explanations into account, sequence annotation would for now be preferable for proteins in the 'no agreement' category. This agrees with our approach in the manuscript in which we primarily give preference to 'legacy' sequence based annotation while supporting them with structural predictions and expand the annotation of those cases in which sequence is not sufficient for functional annotation."

In this context it is important to reiterate that we do not intend for MorF annotations to supersede sequence based annotations.

Minor issue 3

I do not see how the sharing of a Pfam domain between the CoFFE hit and the EggNOG-Mapper hit can be taken as evidence that the two predictions agree.

We agree that overlapping Pfam domains is weak evidence for functional similarity. We decided to add this comparison based on a recent conceptual paper that uses domain content to compare functional equivalence while evaluating structural similarity to detect distant homologs (Monzon *et al.* 2022). We changed the respective sentences to not imply functional similarity in these cases:

“A total of 16,589 proteins were annotated by both MorF and emapper. For 90.6% of these proteins the MorF annotation was homologous to the EggNOG assignment (Fig. 1C), being either orthologs (56.7% in the same metazoan orthologous group) or in the same gene family (33.9% in the same orthologous group at the root level). Proteins that share the same gene family but are not annotated as orthologs either represent paralogs or have been misclassified, a problem for orthology inference that is prone to occur with large evolutionary distances [19]. In the remaining 9.4% of cases approximately half shared a majority of their PFAM domains [20].”

Minor issue 4

Fig 1A - The authors state that the *S. lacustris* structure prediction results in similar average pLDDT scores than other model organisms. However, the median seems quite a bit lower, and superficially one could claim that *S. lacustris* and *D. discoideum* give the lowest scores. Has the NULL hypothesis of equal score distributions been tested?

We thank the reviewer for this observation and useful suggestion. As none of the pLDDT distributions conformed to a Gaussian we performed a two-sided Kolmogorov-Smirnov goodness-of-fit tests between all species pairs with the null hypothesis that their pLDDT distributions are identical. The table below presents the resulting p-values, which shows that all tests reject the null hypothesis, indicating the pLDDT distributions of all species are (statistically significantly) different from each other.

	A_thaliana	M_musculus	D_rerio	S_cerevisiae	H_sapiens	D_discoideum	C_elegans	D_melanogaster	S_lacustris
A_thaliana	1.000000e+00	6.934609e-13	9.114463e-18	7.262317e-19	5.750618e-12	9.016068e-113	4.454955e-08	1.026131e-23	1.648052e-254
M_musculus	6.934609e-13	1.000000e+00	2.070102e-08	5.863687e-16	5.731073e-34	2.383184e-159	5.537467e-06	6.251385e-24	1.329357e-215
D_rerio	9.114463e-18	2.070102e-08	1.000000e+00	4.857841e-19	8.699267e-45	1.086386e-182	1.085280e-13	2.567091e-39	4.784290e-287
S_cerevisiae	7.262317e-19	5.863687e-16	4.857841e-19	1.000000e+00	6.808127e-22	3.284647e-74	1.288430e-22	2.306440e-12	4.903494e-81
H_sapiens	5.750618e-12	5.731073e-34	8.699267e-45	6.808127e-22	1.000000e+00	1.030890e-58	1.705286e-21	8.108620e-11	4.824450e-129
D_discoideum	9.016068e-113	2.383184e-159	1.086386e-182	3.284647e-74	1.030890e-58	1.000000e+00	2.050222e-114	4.987552e-62	3.228598e-15
C_elegans	4.454955e-08	5.537467e-06	1.085280e-13	1.288430e-22	1.705286e-21	2.050222e-114	1.000000e+00	1.466712e-09	1.278533e-145
D_melanogaster	1.026131e-23	6.251385e-24	2.567091e-39	2.306440e-12	8.108620e-11	4.987552e-62	1.466712e-09	1.000000e+00	6.907128e-69
S_lacustris	1.648052e-254	1.329357e-215	4.784290e-287	4.903494e-81	4.824450e-129	3.228598e-15	1.278533e-145	6.907128e-69	1.000000e+00

Based on this, we have now added the following text:

“[...] Average pLDDT values for *Spongilla* predicted protein structures were 4-6 percentage points lower than those of well-characterised animal models (Fig. 1A), likely reflecting the underrepresentation of sponges in the protein structure databases.”

Minor issue 5

The authors state that with their analysis *S. lacustris* has now a functional annotation level that is comparable to *C. elegans*, a widely used model organism. And they refer to Fig. 1B. I wonder whether this is fair to state: First, no CoFFE analysis has been performed with the *C. elegans* proteome. This would likely increase the fraction of annotated proteins. Second, I trust that the functions of many *C. elegans* proteins have experimentally verified, whereas this is not the case for the sponge

We agree and have removed this statement from the manuscript.

Minor issue 6

I am a bit confused about the outcome of the comparison between CoFFE and EggNOG (Fig. 1C). In about a third of the cases the hits belong to the same gene family. What exactly does this mean given that both approaches aim at providing a functional annotation?

Proteins that are labeled to belong to the same gene family are homologous but can not be definitely identified as orthologs, and may instead represent gene paralogs. In practice, this has little effect on the functional annotation of these proteins. By comparing functional annotations provided by the two approaches, the functions of these proteins were identical or highly similar. For instance, 99.9% of MorF and EggNOG mapper based annotations share overlapping GO terms. Nevertheless, careful manual evaluation (e.g. of pLDDT values, bit scores, sequence identity, domain contains, etc.) of both approaches can help assigning the 'best' possible annotation.

Minor issue 7

In Figure 3, the authors present the extension of the ROS metabolism / control based on the CoFFE analysis. I checked only two proteins, FLC3 and FRRS1. FLC3 seems to be a fungal protein, according to orthology databases. It might be that case that its phylogenetic distribution is underestimated due to a rapidly diverging sequence, and it is also present in animals. In the case of FRRS1, however, the situation is different. According to EggNOG, orthologs for this protein are present throughout the eukaryotes http://eggnog5.embl.de/#/app/results?seqid=Q8K385&target_nogs=KOG4293#KOG4293_datamenu. This implies that this gene would have been found also with conventional sequence similarity-based searches. Is this correct? What fraction of the genes marked in red in Fig. 3 are only identified by CoFFE?

We are grateful to the reviewer for pointing out this issue. We reanalyzed the myopeptidocyte marker genes with more care and present the results in Table 5. Blastp and EggNOG 5.0 sequence searches show that many of the MorF annotated proteins don't show any results while others get "rough", protein family wide annotations which are specified by MorF. Figure 3 and the manuscript were updated accordingly. Even after this correction, MorF annotations are central for hypothesizing the ROS metabolism / control function of myopeptidocytes. FLC was not detected using conventional sequence based methods.

Minor issue 8

I suggest to replace the term 'phylogenetic context' with 'phylogenetic profile'.

This has been corrected in the updated version of the manuscript.

Minor issue 9

Instead of using BlastP, it is probably better to search for distantly related / very dissimilar proteins in nrProt using more sensitive approaches, such as Psi-Blast.

We compared MorF to HMMER, one of the most sensitive algorithms for sequence-profile or profile-profile searches. We used eggno-mapper in "hmm" mode and performed sequence-to-profile searches for the translated *S. lacustris* proteome against the eukaryotic HMMs of the EggNOG database. As expected, emapper-hmm is much more sensitive than simple sequence searches, annotating ca. 29,000 proteins (compared to ca. 18,000 by standard emapper and ca. 25,200 by MorF).

The additional sensitivity of emapper-hmm comes at the cost of precision, as ca. 11,700 of the annotations don't have a preferred name, compared to ca. 5,250 for MorF/standard emapper.

Similarly, emapper-hmmer doesn't even produce a description for ca. 2,600 proteins, compared to 38 for MorF and ca. 600 for standard emapper. When accounting for this, MorF still outperforms emapper-hmmer, especially in the cases of cell type marker genes:

“Compared to this combined sequence-based approach, MorF annotates more proteins proteome-wide (~60% to ~40%). More importantly, MorF markedly improved the proportion of annotated cell type- and cell family-specific markers (~70%) compared with sequence-based approaches (~56%; Fig. 1D, Suppl. Fig. S7), even considering sequence profiles (Suppl. Fig. S11)”

A more detailed comparison is presented in the new Supplementary Note I (Suppl. Fig. S10, S11 and Table 4), where we observe broad agreement between annotations where the comparison is possible.

Reviewer 2

My primary concern is that the paper presents this contribution as a pipeline named CoFFE. A pipeline is an implementation of a method (a tool) that can be evaluated on its own and extended to other datasets. But no software is presented. There is nowhere to download and learn to use the pipeline (or I missed it). This makes it difficult to review the methods, and in my mind falls short of the expectations set out in the abstract. The authors should recontextualize their contribution as a method used in this particular analysis of a sponge proteome, rather than an implementation. Or (preferably) make the software available according to standard practices (eg a github repo where others can download the tool to use on their own dataset, with documentation and worked examples) that would be consistent with the presentation of a new "pipeline".

We thank the reviewer for this comment, although we respectfully disagree with their criteria for a pipeline. It is true that the word may evoke a standalone software package or a library for a programming language, which 'CoFFE' (now: 'MorF') isn't. However, we consider "pipeline" a rather appropriate description for MorF, as it is a chain of processing elements arranged so that the output of each element is the input of the next. This is true both in a conceptual level (multiple sequence alignments → structure prediction → structure alignment → functional annotation transfer) and in the implementation level (MMseqs2 → Colabfold/AlphaFold2 → Foldseek → EggNOG-mapper → further post-processing).

There are several reasons that prohibit us from packaging MorF for more convenient use. First and foremost is that most steps of the pipeline cannot be executed on a normal desktop computer. Further, owing to their computational complexity, several parts of the pipeline will need to be adjusted to the user's particular computing environment. These challenges are particularly true for the calculation of multiple sequence alignments (MMseqs2), which requires a lot of processing power (CPU cores) and memory, and depends heavily on the particular input sequences, as well as the prediction of protein structures from these MSAs (colabfold), which requires GPUs with extended memory capabilities (e.g. the Nvidia A100 for longer proteins). Finally, the calculation of structural alignments with Foldseek also profits from having access to many CPU cores and lots of RAM. Hence, the scripts we used can not be general-purpose but rather had to be tailored to the EMBL high-performance compute cluster, where this study was conducted.

We have strived to provide a guide for how users may implement this pipeline on their own particular computing system. For instance, by separating different steps in individual scripts, users have more flexibility in how they implement the different steps, allowing for them to be more easily adapted to different HPC environments (e.g. from slurm-managed environments to LSF or TORQUE ones),

different resource requirements (e.g. for proteomes of different size or more involved alignments) or different tools altogether (e.g. DIAMOND instead of MMseqs2 for MSA generation).

In addition, we have provided extensive documentation for our own implementation of these steps. We made MorF available at <https://git.embl.de/grp-arendt/MorF/> upon submission with a very permissive license, and include usage instructions and documented analysis notebooks that are cited at multiple points in the manuscript. We are happy to report that other groups have already begun using this resource to implement MorF for their own annotation resources.

Lastly, although our pipeline is optimized for large scale predictions and searches (e.g. of a whole proteome), we recognized that users are often interested in only a smaller number of queries, for example of un-annotated differentially expressed genes, or lack the computational resources for a proteome-wide annotation. Therefore, we have now added additional instructions in the Methods section to guide interested readers to the ColabFold server and Foldseek database, where our approach can be reproduced (though not in large scale) without local installation:

"Instructions for MorF searches of single proteins using openly available web servers"

MorF searches for whole proteomes require large computational resources. However, searches can be carried out for a small number of proteins of interest (e.g. top differentially expressed genes in RNAseq or proteomics experiments) using openly available web tools:

- Prediction of protein structure using ColabFold [9]: The structures of proteins of interest can be predicted using the ColabFold Google Colaboratory notebook [93]. Detailed instructions are described in the notebook. For a quick default run, users paste a protein sequence into "query sequence" and hit "Runtime" - "Run all". The results can be downloaded as a zip archive which includes the pdb models of different structure model quality ranks.
- Structure similarity search using Foldseek [7]: The best ranking model ("...rank 1 model X.pdb") can be queried using the Foldseek webserver [94]. Users can upload the pdb file of the model and select databases to use for the search. In default mode, all available databases will be searched. The Foldseek output is structured blast-like and sorted according to best scoring morphologs within the selected databases.
- (Optional) Additional annotation using EggNOG [16]: In order to retrieve additional functional as well as phylogenetic information about the best scoring morpholog, the EggNOG database can be searched [95]. Both protein sequence, as well as UniProt ID can be used to retrieve information about orthologous groups as well as GO-terms, EC numbers, etc."

We hope these changes alleviate the reviewer's concern and present MorF as a pipeline that is usable for both large scale annotation as well as smaller groups of relevant proteins without the need for HPC access. We hope that in the future less computationally intensive alternatives to AlphaFold will allow MorF or similar concepts to be implemented and distributed in a more convenient manner.

page 1 line 54: "protein similarity" -> "protein sequence similarity"

This has been corrected in the updated version of the manuscript.

page 5 line 45: "strengthen the case for" reword. This is a hypothesis you are testing, current wording makes it sound like you already decide this was the case.

We agree with the reviewer and overall weakened the HGT claims in the updated version of the

manuscript, as we currently lack the appropriate sequencing data that would provide more conclusive evidence. We still consider HGT the most likely interpretation of the data, but now discuss alternative interpretations while at the same time stressing MorFs ability to detect genes of non-Metazoan origin (see the amended text below).

page 6 line 1: this is a weak case for HGT without assessing presence in sister groups to animals.

Choanoflagellates are commonly described as the sister group to metazoans. As suggested by the reviewer, we used MMseqs2 to search for the putative HGT candidates in the proteomes of *S. rosetta* and *M. brevicollis*, two emerging models. None of the HGT candidates were detected in their proteomes except for *Spongilla* genes c103983_g1, annotated as an aminohydrolase, which was identified in both organisms². Interestingly, this gene is broadly detected across the (available) sponge phylogeny (see Fig. 4), tentatively placing this transfer event before the split of Choanoflagellates and animals. Conversely, as none of the other candidates were detected, the corresponding HGT events should be posited either in the animal stem or within sponges, such as the proposed freshwater sponge specific HGT candidates.

The amended text can be found in the Results subsection “Polysaccharide hydrolysis in enigmatic mesocytes”:

The absence of these mesocyte-specific hydrolytic enzymes from the digestive toolkit of animals suggests four possibilities for their appearance in the *Spongilla* single-cell data: that they are an artefact (contamination), that they are an evolutionary novelty within *Porifera*, that they were lost in all other animal lineages, or that they were acquired via horizontal gene transfer (HGT).

To explore these different possibilities, we used the marker gene sequences to find putative homologs in the RefSeq non-redundant (nr) [60] and metagenomic databases. Strikingly, the best hits were mostly of bacterial origin, exhibiting 40-70% shared sequence identity with sponge proteins, however mostly lacked annotation (Suppl. Table 1). Notably, we identified putative homologs for each gene in other sponges, suggesting the presence of these sequences in the *Spongilla* protome is unlikely to have occurred due to contamination (Fig. 4B). Consistent with this, we found codon usage and GC content for these genes did not deviate from the *Spongilla* background (Fig. 4A). Lastly, we located all candidate genes on different long contigs (avg. length ~420kb) of an in-house draft assembly of the *Spongilla* genome. The specific co-expression of functionally similar proteins in mesocytes is in contrast to a random contamination.

The prospect of HGT is tantalising. Proteins with enzymatic functions like the ones in the *Spongilla* candidates (polysaccharide hydrolases and metallopeptidases) have been proposed to be horizontally transferred in *A. queenslandica*, a marine demosponge, *S. rosetta*, a choanoflagellate, and *M. leidy*, a ctenophore [61, 62, 63]. Furthermore, *Spongilla* genes c97022_g1 and c103983_g1, a putative aminohydrolase and metallopeptidase respectively, are not only broadly distributed within sponges, but can also be found in the proteomes of choanoflagellates *S. rosetta* and *M. brevicollis* (Suppl. Note L). Furthermore, the *S. rosetta* targets with highest similarity to the *Spongilla* sequences had already been identified as horizontally transferred genes [62]. This would tentatively place this HGT event at least before the split of choanoflagellates and animals (more than ~500mya). Similarly, the phylogenetic distribution of c102757_g1, c95037_g1, and c102838_g2 (yoaJ, sleB, Chitinase class I) would indicate that this group of genes was acquired with the colonisation of freshwater environments (~15-300mya).

² <https://git.embl.de/grp-arendt/MorF/-/blob/main/analysis/revision-hgt-outgroup.ipynb>

Although our analyses suggest these genes may have originated via HGT [64], it is important to consider alternative explanations. One possibility is that these genes are the results of widespread bacterial contamination in sponges genomes and proteomes. Other possibilities include the repeated loss of these genes in other animal lineages, or the convergent evolution of similarity with bacterial proteins. Additional confirmation of their presence in sponge genomes, or evidence of RNA transcripts in sponge cell nuclei would help validate the hypothesis they arose via HGT. Regardless of the source of the genes, MorF annotations provided a novel hypothesis for the elusive function of sponge mesocytes, helping uncover new aspects of sponge biology.

Reviewer 3

Issue 1

All the genome-wide interpretation of new annotations would be much improved by providing information on the level of detail of the annotations. For example CAFA (<https://doi.org/10.1186/s13059-019-1835-8>) uses depth in the Gene Ontology graph. Otherwise it is difficult to compare e.g. the % of genes annotated between nematode and this sponge. If most of the annotations here are "binding" molecular function, it is not informative. The examples which are detailed indicate that annotations can be quite informative, but on the other hand for 1/3 of genes with a known ortholog CoFFE only finds the correct family.

We thank the reviewer for bringing this issue up and giving us the opportunity to clarify and add additional analysis. Based on a similar concern of Reviewer 1 (Minor issue 5), we removed the comparison between sponge and nematode proteome annotation.

Inspired by the reviewer's suggestion, we expanded the comparison between sequence and structure based annotations of the sponge proteome by comparing GO term assignments in different ways (Suppl. Fig. 6) and summarized it in the manuscript:

"We also examined consistency between MorF and sequence-based annotations for Spongilla by comparing GO term overlap, semantic similarity and depth [22], drawing on the annotation comparisons of the CAFA challenge [23]. Almost all (99.9%) proteins annotated by both strategies have at least some overlap in GO terms with 60.6% being identical and 39.3% partially overlapping to various degrees (Suppl. Fig. 6A). GO term semantic similarities between proteins with overlapping (but non-identical) GO annotations additionally reach an average score of 80–90% in the molecular function ontology (Suppl. Fig. 6B)."

Proteins that are labeled to belong to the same gene family are homologous but can not be definitely identified as orthologs, and may instead represent gene paralogs. In practice, this has little effect on the functional annotation of these proteins. By comparing functional annotations provided by the two approaches, the functions of these proteins were identical or highly similar. For instance, 99.9% of MorF and EggNOG mapper based annotations share overlapping GO terms. Nevertheless, careful manual evaluation (e.g. of pLDDT values, bit scores, sequence identity, domain contains, etc.) of both approaches can help assigning the 'best' possible annotation.

We also compared maximum GO term depths between the **overlapping** GO term assignments of sequence and structure annotation pairs of the 'partial overlap' category (Suppl. Fig. 6C). Although this analysis showed that maximum GO term depths on average is between 5-6 for the molecular function ontology, interpretation of GO depth comparisons is tricky. Similar GO term depths in

different branches of the ontology hierarchy can in principle represent different levels of detail. This result is therefore only presented in the Suppl. Figure but not in the main results.

To enhance functional comparisons between sequence and structure based annotations, we also added analysis concerning EC numbers from enzymes which we explain in more detail in the answers to Reviewer 1, Major Issue 1. Again, MorF was able to correctly assign functional annotations across large evolutionary distances, even when homology was not detectable by conventional, sequence-based methods.

Issue 2a

On the one hand, it is clear that a less specific but more sensitive approach would annotate more genes. Thus it would be interesting to compare CoFFE to InterProScan, which like CoFFE has the potential to provide sensitive if not specific annotations.

We compared MorF to HMMER, one of the most sensitive algorithms for sequence-profile or profile-profile searches. We used eggnog-mapper in “hmmer” mode and performed sequence-to-profile searches for the translated *S. lacustris* proteome against the eukaryotic HMMs of the EggNOG database. As expected, emapper-hmmer is much more sensitive than simple sequence searches, annotating ca. 29,000 proteins (compared to ca. 18,000 by standard emapper and ca. 25,200 by MorF).

The additional sensitivity of emapper-hmmer comes at the cost of precision, as ca. 11,700 of the annotations don't have a preferred name, compared to ca. 5,250 for MorF/standard emapper. Similarly, emapper-hmmer doesn't even produce a description for ca. 2,600 proteins, compared to 38 for MorF and ca. 600 for standard emapper. When accounting for this, MorF still outperforms emapper-hmmer, especially in the cases of cell type marker genes:

“Compared to this combined sequence-based approach, MorF annotates more proteins proteome-wide (~60% to ~40%). More importantly, MorF markedly improved the proportion of annotated cell type and cell family-specific markers (~70%) compared with sequence-based approaches (~56%; Fig. 1D, Suppl. Fig. 7), even considering sequence profiles (Suppl. Fig. 11)”

A more detailed comparison is presented in Supp. Note I (Suppl. Fig. 10, 11 and Table 4), where we observe broad agreement between annotations where the comparison is possible.

Issue 2b

On the other hand, since there were many duplications in bilaterian animals since the divergence with sponges, in many cases there will not be one correct ortholog assignment, and EggNOG might be over-classifying. In which case finding a different ortholog might not be a mistake. Thus it would be interesting to either compare to a tool which avoids such over-classification, or to provide some measure of how often such groups of co-orthologs occur. I do not have a specific solution to offer here but at least a discussion of the issue would be welcome.

We completely agree with the reviewer. The incorrect or overclassification of orthologs via sequence-based methods, particularly when considering distantly-related proteins, is well-established (for instance see Natsidis et al. 2021). We would point out that EggNOG is intended to handle cases of many-to-many orthologs. Despite this, nearly all orthology inference algorithms are prone to errors and inappropriately splitting up orthologs proteins into different orthology groups. However, we would note that for the purposes of functional annotation this misclassification does not create a large

problem. Most protein homologs, including both orthologs and paralogs, have very similar functional annotations (e.g. similar GO terms and EC numbers). For most proteins, there is very little information about what distinguishes them functionally. As suggested by the reviewer, we now explicitly discuss this in the main text:

“Proteins that share the same gene family but are not annotated as orthologs either represent paralogs or have been misclassified, a problem for orthology inference that is prone to occur with large evolutionary distances [19].”

Issue 3

For the benchmark of CoFFE on model organisms to be informative relative to the aim of annotating species such as sponge which are distant from model organisms, I would advise to remove from the query database not only the model organisms but their whole lineage (e.g. vertebrates or eudicots).

We thank the reviewer for this suggestion, and Reviewer 3 (issue 3) for raising a similar issue. As the reviewer notes, it is likely that the best non-species hit for any query organism will be from the closest related species in our search databases (except where horizontal gene transfer has occurred).

We revisited Table 3 to exclude species that belong in the same taxonomic group when evaluating the top morphologs. We reanalyzed our comparison of MorF and sequence-based methods in model species by excluding their close relatives, such that the best MorF hit will necessarily be a protein in another animal phylum. Again, we found that the large majority of MorF and sequence-based best hits are homologs. This suggests that structural similarity is indeed a viable method for homology detection even over longer evolutionary distances. We updated the manuscript to reflect this change.

In Results, subsection “MorF annotations agree with sequence-based annotation transfer”:

“We first examined the agreement between morphologs and homologs on available predicted structures of model organisms. We aligned AlphaFoldDB against itself and kept for each query the top morpholog outside the species taxonomic unit. This ensured that MorF would not be identifying quasi identical proteins from closely related species (e.g. *M. musculus* and *R. norvegicus*), simulating a realistic use case where MorF would be used to annotate a non-model organism without well-studied close relatives (Suppl. Table 3). We assessed performance by comparing the eukaryotic orthologous group of the top morpholog to that of the query protein, as defined by the EggNOG v5.0 database. MorF routinely identifies 75-90% of all available homologs, indicating that it can successfully reproduce sequence-based homology inference across large evolutionary distances.”

In Appendix H: Structure-sequence agreement in model species, we updated Table 3:

species	#queries with out-of-group targets	#eligible queries	%eligible agreement	taxonomic group
H. sapiens	20,392	13,838	74.30%	Vertebrata
M. musculus	21,507	15,084	75.05%	Vertebrata
R. norvegicus	19,246	13,180	74.60%	Vertebrata
D. rerio	24,626	13,303	75.36%	Vertebrata
D. melanogaster	13,418	8,828	79.69%	Arthropoda
C. elegans	19,599	11,177	67.29%	Nematoda
S. mansoni	13,845	1,730	91.85%	Nematoda
A. thaliana	27,357	19,800	83.78%	Eudicots
G. max	55,759	32,239	85.40%	Eudicots
Z. mays	38,818	18,644	85.47%	Monocots
O. sativa subsp. japonica	41,794	20,306	86.51%	Monocots
S. cerevisiae (str. ATCC 204508/S288c)	6,013	4,347	83.41%	Saccharomycetales
S. pombe (str. 972/ ATCC 24843)	5,104	3,982	90.51%	Saccharomycetales
C. albicans (str. SC5314/ ATCC MYA-2876)	5,973	4,118	84.41%	Saccharomycetales
C. carrionii	11,169	7,040	81.29%	Chaetothyriales
P. lutzii (str. ATCC MYA-826/Pb01)	8,791	5,086	84.19%	Eurotiomycetes
S. schenckii (str. ATCC 58,251/ de Perez 2211183)	8,652	6,349	78.41%	Sordariomycetes
T. brucei brucei (str. 927/4 GUTat10.1)	8,476	4,303	60.26%	Trypanosomatida
T. cruzi (str. CL Brener)	19,026	8,782	56.56%	Trypanosomatida
L. infantum	7,914	4,423	56.14%	Trypanosomatida
D. discoideum	12,584	6,312	63.96%	Amoebozoa

It now lists the broad taxonomic group to which we assigned each species. These reflect clades, orders, phyla, or even kingdoms, depending on the prevalence of related organisms in the database. We updated the accompanying text as well:

“As proof of principle we explored the power of structural similarity to identify orthologous genes for a number of model species whose proteomes already had predicted structures in AlphaFoldDB. We aligned AlphaFoldDB against itself using Foldseek. For each query protein we kept the best target from outside the taxonomic group and compared the eukaryotic orthogroup assignment. The best structural hit overwhelmingly belonged to the same orthogroup as the query (Suppl. Table 3).

We used the Proteins API [112] to obtain the species name for each entry [113]. We decorated each entry with its orthogroup assignment from the EggNOG database [17] (v5.0). Not all proteins currently in AlphaFoldDB could be assigned an orthogroup; we ignored comparisons where the query or its best target were missing the orthogroup assignment. The full table can be found in the corresponding notebook [114].

We compared the remaining eligible cases and found overwhelming agreement between the eukaryotic orthogroup of the best structural target and the orthogroup of the query protein, confirming that structural similarity can, in principle, detect homology relationships. Though the available targets were restricted to be outside the clade, order, phylum, or even kingdom, top morphologs still are homologs in the large majority of cases, a very encouraging indication that MorF is able to detect homology over long evolutionary distances.”

Spongilla lacustris is only distantly related to animal models and is as such the intended use case for our pipeline. For *Spongilla* proteins the best hit is always a protein from a different animal phyla. We show MorF consistently identifies cross-phyla homologs in ~90% of cases, further confirming what Table 3 suggests, namely that structural similarity is well-suited for homology detection across long evolutionary distances.

"Spongilla lacustris, an early-branching animal": What does early-branching mean here? I think the main point in context is that has diverged very long ago from any well annotated model organism.

This was not clearly formulated and has been updated in the new version of the manuscript:

"Sponges (*Porifera*) are animals that diverged early in the Metazoan tree relative to well annotated model organisms such as human and mouse"

I don't understand what the following sentence means: "Considering that careful evaluation of sequence-based phylogenetic trees can identify a protein's evolutionary history [55], it will be interesting to explore the extent to which annotations based on structure provide better predictors of function than evolutionary homology."

We look forward to future experimental efforts that will try to answer the question whether a protein's evolutionary ancestry (homology), or its closest structural hits are better indicators of the 'true' function of a protein.

We rephrased the ambiguous sentence in the updated manuscript:

"In the future, it will be interesting to explore these cases experimentally to test whether structural similarity or homology is a better predictor of protein function [66]."

*"Relative alignment length (FS) (percentage of the query (*Spongilla*) structure aligned with the best target structure) does not correlate too strongly": Do you mean "correlates weakly"?*

We changed the sentence to the proposed formulation. Positive formulations are better!

I found ref 8 was now published at <https://www.nature.com/articles/s41592-022-01488-1>.

We changed the citation accordingly.

Availability of data and materials: the link to zenodo is incorrect.

We replaced the zenodo DOI with the full URL to the zenodo repository.

Bibliography

[1] Zhou N, Jiang Y, Bergquist TR, Lee AJ, Kacsoh BZ, Crocker AW, Lewis KA, Georghiou G, Nguyen HN, Hamid MN, Davis L. The CAFA challenge reports improved protein function prediction and new functional annotations for hundreds of genes through experimental screens. *Genome biology*. 2019 Dec;20(1):1-23.

[2] Kohn T, Wiegand S, Boedecker C, Rast P, Heuer A, Jetten MS, Schüler M, Becker S, Rohde C, Müller RW, Brümmer F. *Planctopirus ephydatiae*, a novel Planctomycete isolated from a freshwater sponge. *Systematic and applied microbiology*. 2020 Jan 1;43(1):126022.

[3] Monzon V, Paysan-Lafosse T, Wood V, Bateman A. Reciprocal best structure hits: using AlphaFold models to discover distant homologues. *Bioinformatics Advances*. 2022;2(1):vbac072.

Second round of review

Reviewer 1

Review of the revised version „Beyond sequence similarity: cross-phyla protein annotation by structural prediction and alignment” submitted by Ruperti et al.

Having now read the revised version of this manuscript, I think that the authors were overall very carefull in addressing the comments raised by all three reviewers. However, few questions remain open, which I list below.

Major issues

1.

a. I regret to say that I am quite confused about the way how the terms ‘orthology’, ‘paralogy’ and ‘homology’ are used in this manuscript. To give an example, on page 3, line 43 the authors write “Traditional approaches for functional annotation use protein sequences to identify orthologous or homologous proteins.”. It might be semantically only a tiny issue, but the ‘or’ implies that ‘homologous proteins’ are something different that ‘orthologous proteins’, which of course is not true, since orthologs are are a subset of homologs. In this case, the issue can be simply resolved by writing “Traditional approaches for functional annotation use protein sequences to identify orthologous or, more generally, homologous proteins.”. However, more generally, I have the impression that the authors seem to wobble between ‘orthologs’ and ‘homologs’. The manuscript would become significantly stronger if the authors would find a clear line here.

b. Along the same lines as (1), the statement “(...) and effectively treat functional annotation synonymously with assessment of homology.” is an oversimplification that does not read well in the context of this manuscript. The community is well aware that homology is only one indicator for a functional annotation transfer, the authors are of course aware of the plethora of evidence codes within the GO alone, and then people typically call it a ‘tentative annotation’ at most. I strongly suggest to describe protein sequence similarity-based approaches for a functional annotation transfer more carefully.

My personal(!) thoughts are the following; maybe the key sentence in the background is not helpful. The authors write “Here we propose a pipeline for protein annotation using structural similarity, exploiting the fact that similar protein structures often reflect homology and are more conserved than protein sequences.”. This sentence ties the entire manuscript to the concept of annotation transfer based on sequence homology, with hundreds of papers, many of them of high relevance, that precisely discuss what should be and what shouldn’t be done during annotation transfer. I wonder now if the authors should not better place the emphasis on the connection of a significant structural similarity as a proxy for functional equivalence, and then use amino acid sequence similarity-based approaches to identify orthologs, or more generally homologs, to benchmark this approach. In the discussion one could then briefly discuss why two proteins are structurally (and functionally) similar: (i) they share a common evolutionary origin, which we sometimes can and sometimes cannot infer based on sequence similarity or (ii) the structures emerged convergently. Once again, these are just my thoughts,

and the authors are free to consider them or not. In the latter case, however, 1a and 1b should be addressed.

2. In their performance assessment of MorF, the authors investigate yeast enzymes with no detectable ortholog in humans. This analysis shows that the detected protein pairs with significant structural similarity show a large overlap in their EC annotation. What are the authors ideas about the evolutionary relationships of the yeast and human proteins given that yeast enzymes are very benign when it comes to identifying remote homologs (see <https://doi.org/10.1093/gbe/evz008>). In any case, the corresponding protein pairs (also for the human – Arabidopsis comparison) should be provided as a supplementary table.

Minor issues

1. The use of the term ‘legacy’ in the context of sequence similarity-based approaches for functional annotation transfer suggests that this approach is outdated. I do not believe that we are that far at the moment.
2. yoaJ is referred to as a new annotation based on MorF, however in Table 1 it is referred to as ‘refined by MorF’. I think I could not find an explanation for the nature of this ‘refinement by MorF’, so maybe this can be explained.
3. Page 3, line 54 – taxonomic unit is a too general term to leave it unexplained in the text.
4. Page 3, line 63 – the benchmark shows that up to 25% of the homologs are missed by MorF. I do not want to be overly negative, but missing a quarter is probably too much to make the statement “(...) indicating that it can successfully reproduce sequence-based homology inference across large evolutionary distances” a valid statement. Moreover, what does ‘routinely’ mean in the context of a MorF analysis?
5. The sentence “It is important to note that it is impossible to know for sure whether or not these represent homologs or protein structures that evolved via convergent similarity.” should be rephrased.
6. Reference 22 is published by now

Reviewer 2

All major concerns have been adequately addressed. I find this a fascinating paper that I am sure will be of interest to many.

Response to Referees: Ruperti, Papadopoulos et al.

Introduction

We thank the reviewers for their in-depth review which has significantly improved our manuscript. We have addressed all remaining issues and in particular agree on the necessity to further distinguish the usage of homolog vs. ortholog/paralog.

Please find our answers to the reviewer's comments and suggestions below. Text passages that are copied verbatim from the manuscript are highlighted in blue.

Reviewer 1

Major issue 1

1a

I regret to say that I am quite confused about the way how the terms 'orthology', 'paralogy' and 'homology' are used in this manuscript. To give an example, on page 3, line 43 the authors write "Traditional approaches for functional annotation use protein sequences to identify orthologous or homologous proteins.". It might be semantically only a tiny issue, but the 'or' implies that 'homologous proteins' are something different than 'orthologous proteins', which of course is not true, since orthologs are a subset of homologs. In this case, the issue can be simply resolved by writing "Traditional approaches for functional annotation use protein sequences to identify orthologous or, more generally, homologous proteins.". However, more generally, I have the impression that the authors seem to wobble between 'orthologs' and 'homologs'. The manuscript would become significantly stronger if the authors would find a clear line here.

We agree with the reviewer that the terms were used inconsistently at several points in the manuscript. We changed the paragraph accordingly, also taking point 1b) into account. In the revised manuscript, we now use 'homology' when both orthology and paralogy are necessary to consider, and reserve "ortholog" and "paralog" for only cases where a distinction can be made.

In Subsection "MorF annotations agree with sequence-based annotation transfer", we rephrased the first paragraph:

"Experimental evidence for the function of a protein is only available for a vanishingly small number of known protein sequences. For the rest, annotations are propagated, mostly using sufficient sequence similarity as a proxy for homology. In particular, orthology has often been used to transfer functional annotation wholesale, even though it is known that divergence of function within orthologs is possible. In the absence of robust high-throughput alternatives, sequence-based homology detection remains the standard for functional annotation. To assess MorF annotations, we compare them to those produced via EggNOG v5.0 and EggNOG-mapper, representing the state-of-the-art in homology detection".

In another case we changed 'orthology' to 'homology' where a more general usage of the term was appropriate:

"In non-model species, the sequence-based prediction of gene homology can be used to infer protein identity, however this approach loses predictive power with longer evolutionary distances."

1b

Along the same lines as (1), the statement "(...) and effectively treat functional annotation synonymously with assessment of homology." is an oversimplification that does not read well in the context of this manuscript. The community is well aware that homology is only one indicator for a functional annotation transfer, the authors are of course aware of the plethora of evidence codes within the GO alone, and then people typically call it a 'tentative annotation' at most. I strongly suggest to describe protein sequence similarity-based approaches for a functional annotation transfer more carefully.

We updated the paragraph to offer a more nuanced description of annotation transfer using sequence similarity and inferred homology, see the subsection "MorF annotations agree with sequence-based annotation transfer" above.

My personal(!) thoughts are the following; maybe the key sentence in the background is not helpful. The authors write "Here we propose a pipeline for protein annotation using structural similarity, exploiting the fact that similar protein structures often reflect homology and are more conserved than protein sequences.". This sentence ties the entire manuscript to the concept of annotation transfer based on sequence homology, with hundreds of papers, many of them of high relevance, that precisely discuss what should be and what shouldn't be done during annotation transfer. I wonder now if the authors should not better place the emphasis on the connection of a significant structural similarity as a proxy for functional equivalence, and then use amino acid sequence similarity-based approaches to identify orthologs, or more generally homologs, to benchmark this approach. In the discussion one could then briefly discuss why two proteins are structurally (and functionally) similar: (i) they share a common evolutionary origin, which we sometimes can and sometimes cannot infer based on sequence similarity or (ii) the structures emerged convergently. Once again, these are just my thoughts, and the authors are free to consider them or not. In the latter case, however, 1a and 1b should be addressed.

We share similar thoughts with the reviewer, and explore different facets of the relationships between sequence, structure, and function in our manuscript. We probed the agreement between structural and sequence similarity in multiple scenarios, and observed that top morphologs were consistently homologous according to their sequences. Ultimately, this led us to believe that homology is the main and most common reason for structural similarity and functional conservation. We thus decided to frame MorF as a more sensitive homology detector, rather than as a functional equivalence detector. At the same time, it is true that convergence cannot be excluded as a probable reason for structural similarity, so we expanded and rephrased the discussion to include the reviewer's suggestion:

"The fact that morphologs are overwhelmingly homologs shows that structural and functional similarity between proteins mostly results from common descent. However, similar structures may, and do, also emerge convergently. MorF and sequence-based annotations disagree in 5% of cases, representing either technical artifacts, homology we simply cannot infer, or examples of convergent evolution. In the future, it will be interesting to explore experimentally whether structural similarity or homology is a better predictor of protein function."

Major issue 2

In their performance assessment of MorF, the authors investigate yeast enzymes with no detectable ortholog in humans. This analysis shows that the detected protein pairs with significant structural similarity show a large overlap in their EC annotation. What are the authors' ideas about the evolutionary relationships of the yeast and human proteins given that yeast enzymes are very benign when it comes to identifying remote homologs (see <https://doi.org/10.1093/gbe/evz008>). In any case, the

corresponding protein pairs (also for the human - *Arabidopsis* comparison) should be provided as a supplementary table.

As described in Major issue 1b, it is possible that these proteins are homologs with a common origin only detectable based on structure similarity. An alternative is that these proteins are not related, yet have evolved similar functions convergently. At this point we cannot exclude either possibility.

We provide a jupyter notebook that details how the table is produced (https://git.embl.de/grp-arendt/MorF/-/blob/main/analysis/revision-remote_species.ipynb). We added the tables of human/yeast and human/*Arabidopsis* morphologs, including EggNOG orthologous group information, preferred names, and orthology/homology assessment, to the zenodo repository since they were too large to add as supplementary tables.

Minor issues

1

The use of the term 'legacy' in the context of sequence similarity-based approaches for functional annotation transfer suggests that this approach is outdated. I do not believe that we are that far at the moment.

We agree that the chosen term might imply an outdated approach. We amended the text in different parts of the manuscript to explain that we use “legacy annotation” as shorthand for the annotation produced by Musser *et al.* in the *Spongilla* single cell atlas, and tried to emphasise that we still consider sequence similarity the more reliable predictor of homology:

Results, “A protein structure-based workflow enriches functional annotation for *Spongilla lacustris*”: We also compared our results to annotations from the recently published *Spongilla* cell type atlas, which used BLASTp to supplement emapper annotations. **We refer to these as “legacy annotations”.**

Results, Fig. 2:

Genes on blue background were **annotated by Musser *et al.* with sequence-based methods (“legacy annotation”)**. Genes on yellow background are annotated by MorF.

Materials and methods, “Legacy annotation”:

Musser *et al.* used the putative proteins to create a phylome by constructing gene/protein trees for each protein. The phylome information was used to refine the assignment of transcripts to genes. In some cases, 3' and 5' fragments of a gene were assigned to two different transcripts. These fragments were merged into the same merged gene name using the gene tree information. Functional annotations were supplemented by EggNOG mapper (v1) and *blastp* searches against human RefSeq (default parameters). This annotation was used in the original *Spongilla* scRNA sequencing publication and is present in the single-cell data. **We refer to this as “legacy annotation”.**

Suppl. Material, “Comparison of MorF parameters between different agreement categories of structure and sequence based annotations”

Taking the possible explanations into account, sequence annotation would for now be preferable for proteins in the “no agreement” category. This agrees with our approach in the manuscript in which **we primarily give preference to “legacy” sequence based annotation** while supporting them with structural predictions and expand the annotation of those cases in which sequence is not sufficient for functional annotation.

2

yoaJ is referred to as a new annotation based on MorF, however in Table 1 it is referred to as 'refined by MorF'. I think I could not find an explanation for the nature of this 'refinement by MorF', so maybe this can be explained.

'Refined by MorF' has not been previously defined in the manuscript. In these cases, emapper did not return a 'preferred name' or 'preferred description', which we define to be a requirement for 'annotation'. However, "legacy annotations" from Musser *et al.*, which include the phylome, emapper as well as blastp, sometimes returned sparse information such as single domains for these proteins. To clarify this, we updated the corresponding paragraph:

"New and refined annotations provided by MorF include proteins such as expansin (*yoaJ*), glucan endo-1,3-beta-glucosidase (BG3), and spore cortex-lytic enzyme (*sleB*), all hydrolases that specifically degrade cell walls, cellulase, chitin, other polysaccharides and proteins. **"Refinement" here indicates that MorF provided names or descriptions for proteins previously annotated more sparsely, e.g. by a single predicted domain."**

3

Page 3, line 54 - taxonomic unit is a too general term to leave it unexplained in the text.

We agree that 'taxonomic unit' was not specified clearly enough. We added this information in the main results:

"We aligned AlphaFoldDB against itself and kept for each query the top morpholog outside the species taxonomic unit. For Metazoa, we excluded all species belonging to the same phylum. Viridiplantae were divided into monocots and eudicots whereas fungi and trypanosome species were grouped by class."

4

Page 3, line 63 - the benchmark shows that up to 25% of the homologs are missed by MorF. I do not want to be overly negative, but missing a quarter is probably too much to make the statement "(...) indicating that it can successfully reproduce sequence-based homology inference across large evolutionary distances" a valid statement. Moreover, what does 'routinely' mean in the context of a MorF analysis?

We rephrased the sentence and removed the word 'routinely' to better reflect the results of the benchmark:

"MorF identifies 75-90% of all available homologs, indicating that it is largely able to reproduce sequence-based homology inference even across long evolutionary distances."

5

The sentence "It is important to note that it is impossible to know for sure whether or not these represent homologs or protein structures that evolved via convergent similarity." should be rephrased.

We rephrased the sentence in the updated manuscript:

"However, it is important to note that these cases could either represent homologs or protein structures that evolved convergently."

6

Reference 22 is published by now

We updated the bibliography accordingly.

7

Following up Ref. 2's question about the availability of the MorF pipeline. From the response I understand that a downloadable pipeline does not exist. Maybe, to avoid any further confusions, it is worthwhile referring to MorF as a "workflow" instead of a "pipeline".

'Workflow' indeed is a better descriptor of MorF. We changed the wording accordingly in the updated manuscript.